# Spectral-temporal-spatial customization via modulating multimodal nonlinear pulse propagation

Tong Qiu[1,3], Honghao Cao[1,3], Kunzan Liu [1], Li-Yu Yu [1], Manuel Levy[2], Eva Lendaro[2], Fan Wang [2] & Sixian You [1] ✉

Multimode fibers (MMFs) are gaining renewed interest for nonlinear effects due to their high-dimensional spatiotemporal nonlinear dynamics and scalability for high power. High-brightness MMF sources with effective control of the nonlinear processes would offer possibilities in many areas from high-power fiber lasers, to bioimaging and chemical sensing, and to intriguing physics phenomena. Here we present a simple yet effective way of controlling nonlinear effects at high peak power levels. This is achieved by leveraging not only the spatial but also the temporal degrees of freedom during multimodal nonlinear pulse propagation in step-index MMFs, using a programmable fiber shaper that introduces time-dependent disorders. We achieve high tunability in MMF output fields, resulting in a broadband high-peak-power source. Its potential as a nonlinear imaging source is further demonstrated through widely tunable two-photon and three-photon microscopy. These demonstrations provide possibilities for technology advances in nonlinear optics, bioimaging, spectroscopy, optical computing, and material processing.

Multimode fibers (MMFs) have become renewed focuses of interest for investigating multimode nonlinear optics, owing to their versatile degrees of freedom for guided waves and their potential for high power scalability[1–12]. Over the last 5–10 years, significant progress has been made in studying and controlling nonlinear effects in MMFs, presenting great opportunities for a plethora of applications in optical sensing, imaging, manipulation, and computing. For example, the application of MMFs to fiber lasers could permit the development of low-cost light sources with dramatically higher pulse energy and average power due to the larger mode areas of the MMFs[13–16]. More importantly, the rich spatiotemporal dynamics and complex inter-modal interactions in MMFs constitute a broad avenue for controlling nonlinear wave propagation, opening up possibilities for intriguing physics and applications such as spatiotemporal light control[17,18], nonlinear frequency generation[4,12,19–21], nonlinear optical imaging and sensing[22–25], optical wave turbulence[26–29], and optical computing[30,31].

Recently, control of the multimode spatiotemporal light fields has been demonstrated by adjusting the input wavefront to graded-index (GRIN) MMFs for selective mode excitation[11,15,32–36]. These studies, by exploiting the spatial degree of control of the input field, demonstrated the great potential of MMFs as technological solutions for tunable high-power light sources and spatiotemporal avenues for fundamental nonlinear optical physics. However, the primary focus on spatial control in GRIN MMFs often subjects the spatiotemporal pulses to limited capabilities in many applications due to two major problems: difficulty in control and limited broadband spectral brightness. Fiber sources with higher-dimensional tunability and broadband spectral brightness could usher in a wave of opportunities for light sources and nonlinear phenomena[9,37–43], and ignite advances in cutting-edge applications, such as bioimaging, laser manufacturing, chemical sensing, and optical computing. To realize this potential, harnessing the untapped potential of temporal degrees of freedom to

[1]Department of Electrical Engineering and Computer Science, Massachusetts Institute of Technology, Cambridge, MA, USA. [2]Department of Brain and Cognitive Sciences, Massachusetts Institute of Technology, Cambridge, MA, USA. [3]These authors contributed equally: Tong Qiu, Honghao Cao. ✉e-mail: sixian@mit.edu

control nonlinear effects in MMFs at high peak power levels presents a compelling solution.

Here we introduce a simple but effective way of controlling multimodal nonlinear effects in high-power regimes by exploiting not only the spatial but also the temporal degrees of freedom. Complementary to manipulating the input pulses that seed the nonlinear pulse propagation, we propose to modulate the multimodal nonlinear pulse propagation processes by introducing programmable time-dependent disorders along the fiber[26], which is implemented as axial-position-dependent macro-bending. This was made possible by using a single 3D-printed modulating device dubbed fiber shaper (conceptually depicted in Fig. 1a). By applying the fiber shaper to a standard silica step-index (SI) fiber at high-power levels (Fig. 1b), we showed effective modulation of the multimodal nonlinear pulse propagation and customization of the MMF output field in the spectral, temporal, and spatial domains.

We further demonstrated the ability to generate high peak power even extending outside the conventional soliton regime. The broadband high peak power (approaching megawatt on average) was accomplished by combining spectral energy reallocation (up to 166-fold) and temporal shortening (up to 4-fold). Such capabilities are uniquely enabled by the fiber shaper, a single all-fiber modulator without the need for external wavefront or pulse shaping. To showcase its potential, we applied the apparatus directly to multiphoton microscopy, an application with demanding needs for spectral and temporal properties of pulses. Through adaptive optimization of the fiber shaper, we achieved efficient and widely tunable two-photon and three-photon microscopy of fluorescent beads and label-free tissues. Our proposed approach, enabling simultaneous access to the spatial and temporal degrees of control of multimodal nonlinear effects, relies on the application of precisely controlled multi-point macro-bending to the fiber. Recent demonstrations have shown the use of

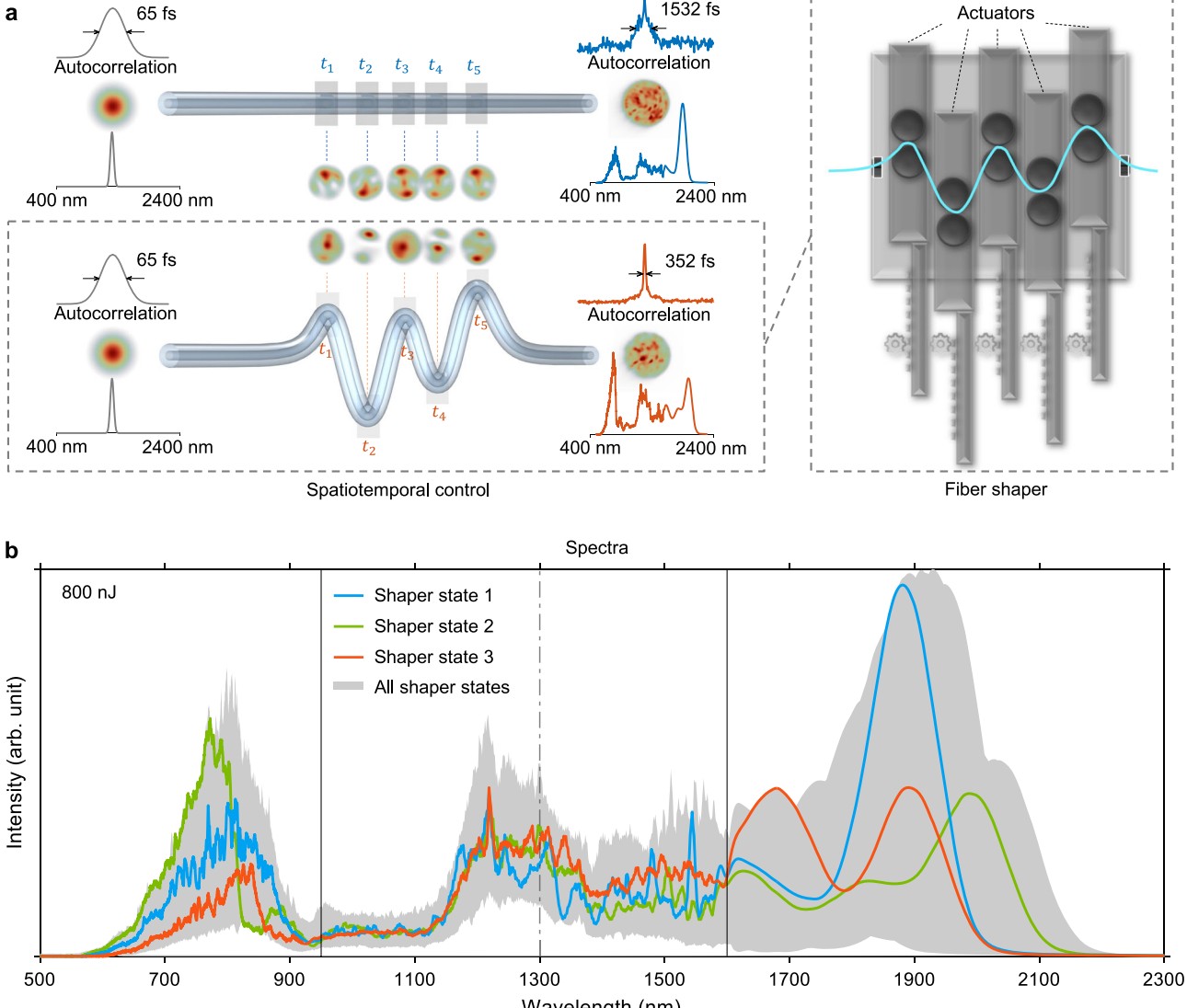

**Fig. 1 | Conceptual demonstration of spatiotemporal control of nonlinear pulse propagation in MMFs by a fiber shaper. a** Illustration of the principle of spatiotemporal control of multimodal nonlinear effects in MMFs by introducing programmable time-dependent disorders using a fiber shaper. The fiber shaper applies multi-point macro-bending of various bending radii to the fiber and thus alters the multimodal spatial interactions at different time points (e.g., $t_1$ to $t_5$) along the pulse temporal evolution. **b** Visualization of experimentally acquired output spectra range of an SI MMF with the same launching condition but different combinations of actuator positions. A total of 3125 configurations were exhaustively searched, involving 5 actuators, each with 5 different states. Three representative spectra corresponding to three randomly chosen configurations are highlighted in distinct colors. Vertical solid lines mark the spectral range measured by different spectrometers. The dash-dotted line denotes the input wavelength. Source data are provided as a Source Data file.

piezoelectric actuators (i.e., fiber pianos) to induce controllable bends in MMFs for modulating modal dynamics in the linear regime[44–47]. Inspired by this design, we want to develop a module that's compatible with nonlinear regime and overcomes the challenge of tunability and low spectral brightness in fiber sources. Despite the promising applications in linear and quantum regimes[46–48], it is challenging to directly adopt linear fiber pianos to the nonlinear regime due to application-specific issues such as light loss, design flexibility, and scalability. By employing the slip-on fiber shaper created through 3D printing that supports smooth (minimal micro-bending) but large-curvature bending (sufficient model coupling effects), we envision its potential as an accessible and flexible tool for both linear and nonlinear modulations of MMFs, with significant implications in nonlinear optics, bioimaging, and spectroscopy.

## Results

### Experimental setup

Ultrashort pulses (Light Conversion Cronus-3P) were launched into a standard silica SI MMF with a length of 30 cm by weakly focusing (see Methods). To explore the fiber shaper's capability in both the normal and anomalous dispersion regimes, we initiated the experiments with a pump wavelength of 1300 nm (46 fs). Due to its proximity to the zero-dispersion wavelength of silica, a wide array of nonlinear effects can be effectively excited, including self-phase modulation (SPM), cross-phase modulation (XPM), four-wave mixing (FWM), multimode soliton formation, and dispersive wave emission[49–52]. The fiber was mounted on a 3D-printed programmable fiber shaper that introduces multi-point macro-bending to the MMF through five individually controlled motorized actuators (Fig. 2). Each actuator locally and precisely applies controlled bending on the fiber at multiple axially dispersed positions. By producing local index perturbations, the actuators cause energy coupling between modes[53,54] at multiple time points of the pulse evolution. When operated together, these actuators exert the high-dimensional spatiotemporal control of the multimodal nonlinear pulse propagation, thus creating opportunities for simultaneous customization of the spectral, spatial, and temporal profiles of the fiber output pulses. Besides the ease of construction and operation, the fiber shaper maintains high light throughput with negligible transmission loss regardless of the actuator configurations, measured at ±1% at high power levels (input pulse energy of 800 nJ, 85% coupling efficiency at initial states).

To understand the multimodal compositions of the spatiotemporal nonlinear effects in the SI MMF, the fiber output was spatially profiled with multiple spectral bands of interest (see Supplementary Fig. 1). The wavelength-dependent spatial profiles of the fiber output indicate that the lower-order modes are responsible for the formation of the most red-shifted multimode soliton (1900 ± 100 nm; Supplementary Fig. 1b), whereas the generation of the dispersive waves (560 ± 5 nm; 725 ± 25 nm; Supplementary Fig. 1b) is dominated by the higher-order radially symmetric modes, as predicted in[50], to satisfy the intermodal phase-matching condition between one of the modes comprising the dispersive waves and another lower-order mode of the multimode soliton. The region in between (i.e., SPM, XPM, and FWM) (900 ± 25 nm, 1200 ± 5 nm; Supplementary Fig. 1b) exhibits a high degree of multimodal behavior, which could be attributed to the many degrees of freedom required to satisfy the intermodal group velocity matching and intermodal phase matching[20,55,56]. These mechanistic studies and observations not only shed light on the underlying mechanisms of the two-octave spectral broadening, but also highlight the multimodal compositions of the spectral broadening in different dispersion regimes, which lays the foundation for the broadband tunability discussed in the following sections.

### Effective spatiotemporal control of multimodal nonlinear effects in SI MMF

To achieve flexible tunability through spatiotemporal control of the multimodal nonlinear effects, we designed a fiber shaper consisting of five 3D-printed linear actuators (Fig. 3a) at a total cost of 35 US dollars including the motors and its actuators (see Methods, Supplementary Note 2, and Supplementary Figs. 4–6). The actuators can independently translate in the direction perpendicular to the optical axis. By controlling the relative shift between the actuators, macro-bending of various radii can be applied to the fiber precisely and simultaneously, allowing for an exhaustive automated search and adaptive optimization of thousands of fiber shape configurations to achieve optimal output spectral-temporal-spatial properties. This can be readily scaled up with more actuators to unlock the full high-dimensional spatiotemporal control of the multimodal nonlinear optics (see Supplementary Fig. 9). In principle, these actuators alter the local refractive index profile, causing energy coupling between modes at multiple time points during the pulse evolution, which altogether enable the high-dimensional spatiotemporal control of the multimodal nonlinear pulse propagation. This process can be modeled by supplementing the generalized multimode nonlinear Schrödinger equation (GMMNLSE) (see Methods) on the right-hand side with an additional term representing the linear mode coupling effect[55,57,58]:

$$i \sum_{n}^{N} Q_{np} A_n(z,t), \tag{1}$$

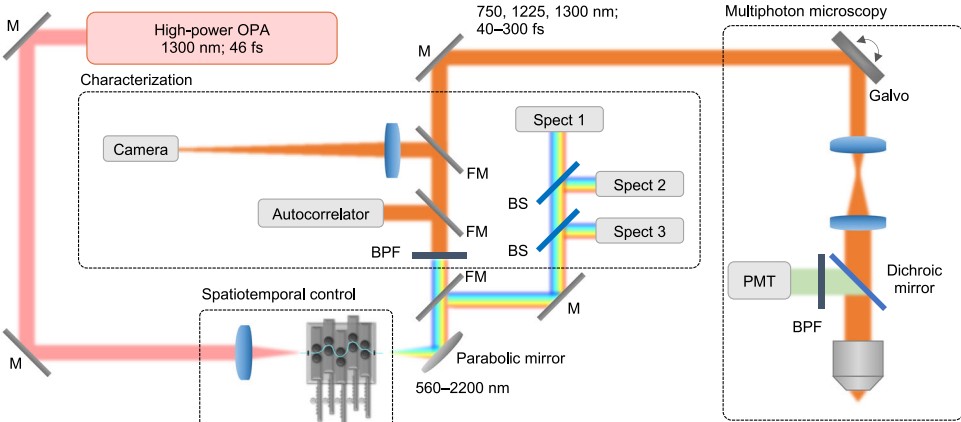

**Fig. 2 | Schematic of the experimental setup.** The ultrabroadband light out of the SI MMF is collimated by an off-axis parabolic mirror before directing to the characterization apparatus and imaging system (see Methods for details). OPA optical parametric amplifier, M mirror, FM flip-mirror, BPF bandpass filter, BS beam splitter, Spect spectrometer, PMT photomultiplier tube.

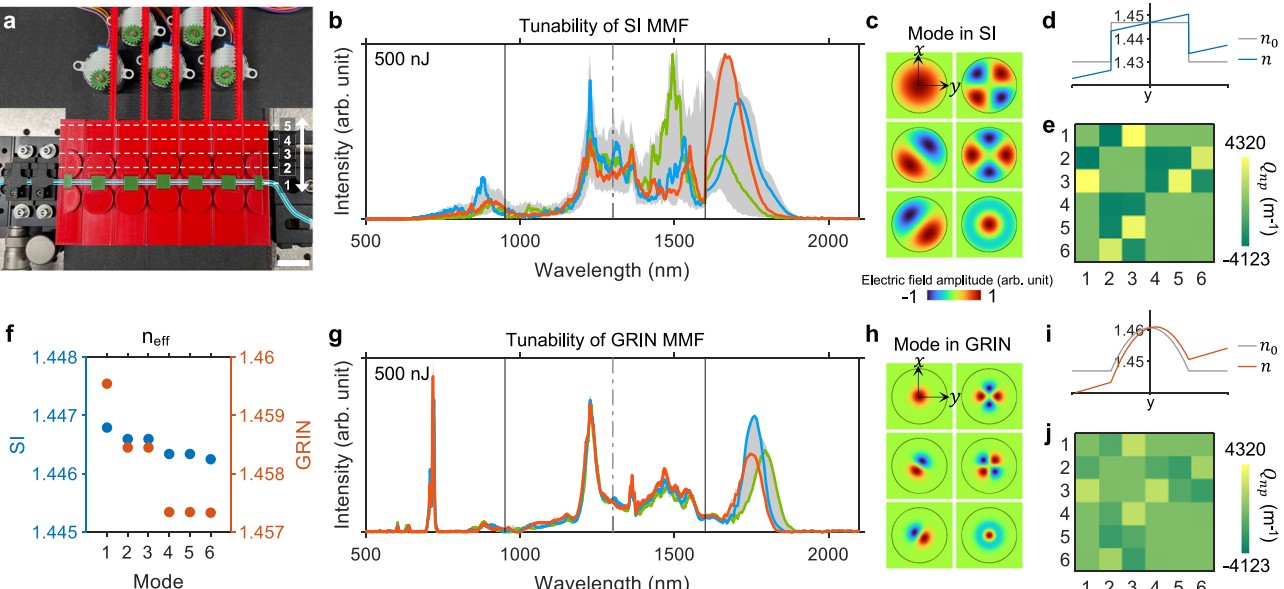

**Fig. 3 | Mechanisms of spatiotemporal control of nonlinear effects in SI MMFs using fiber shaper. a** Photograph of the custom-designed fiber shaper at its initial state. Scale bar: 25.4 mm. Experimental results of 30-cm-long SI MMF (50/125 μm, 0.22 NA) (**b**) and GRIN MMF (50/125 μm, 0.2 NA) (**g**) with the same set of macro-bending applied. Representative spectra corresponding to three randomly chosen configurations out of a pool of 3125 configurations are highlighted in distinct colors. The input pulse energy of 500 nJ is experimentally limited by the laser-induced damage to the GRIN MMF. Normalized electric field of the first six spatial modes in unperturbed SI MMF (**c**) and GRIN MMF (**h**), with the core-cladding interface marked in black. Illustrative examples of the refractive index profiles of straight ($n_0$) and curved ($n$) SI MMF (**d**) and GRIN MMF (**i**). $n$ is approximately expressed by $n_0 + n_0 y/r_b$, where $r_b$ represents the bend radius, which is set to 1 cm (the minimum value the fiber shaper can introduce by design). Linear mode coupling coefficient $Q_{np}$ resulting from the macro-bending in **d**, **i** for the first six spatial modes in **c**, **h** in SI MMF (**e**) and GRIN MMF (**j**). **f** Effective refractive index ($n_{eff}$) for the first six spatial modes in SI and GRIN MMFs. SI: step-index; GRIN: graded-index. Source data are provided as a Source Data file.

where $Q_{np}$ denotes the linear coupling coefficient between mode-$n$ and mode-$p$ due to a local perturbation, determined by the spatial overlap of the three quantities (see Methods); $A_n(z,t)$ represents the temporal envelope of mode-$n$, which is complex-valued to include the envelope phase due to distinct phase velocity of each mode[58,59]; $z$ is the propagation direction. The modes can be simplified as the ideal modes of the unperturbed waveguide using perturbative coupled-mode theory in principle[60]. However, in practice, the mode field can undergo considerable deformation in curved multimode fibers, making it necessary to consider additional variations related to mode field deformation. These variations include changes in the propagation constant and in the nonlinear coupling coefficient $S_{plmn}^K$ and $S_{plmn}^R$ (see Methods), which are determined by the spatial overlap between the mode-$p$, -$l$, -$m$, and -$n$. As a result, by introducing localized changes to $Q_{np}$ and to the mode fields at axially dispersed positions along the fiber, the spatial and temporal degrees of freedom of nonlinear pulse propagation can be simultaneously controlled.

The perturbed GMMNLSE, along with the observed evolution of the spectral broadening and spectrally distinct spatial profiles (Supplementary Figs. 1 and 3), provide a key insight into the condition of effective spatiotemporal control of multimodal nonlinear optical processes in multimode fibers. At individual temporal instances, the more effectively we can modulate the linear and nonlinear mode coupling coefficients and the mode propagation constants, the higher-dimensional control and the greater tunability the source will exhibit. A lot of previous studies on multimodal nonlinear pulse propagation focused on GRIN MMFs due to their unique self-imaging properties and low modal dispersion. However, we observed that SI MMFs exhibit significantly greater tunability and higher spectral brightness under the same experimental conditions (Fig. 3b, g). This is likely due to their larger modal areas and the more closely spaced propagation constants, which collectively lead to lower peak intensity, higher sensitivity to bending, and increased susceptibility to the mode coupling effect.

To gain a mechanistic understanding of this phenomenon, we simulated the effect of bending for the two fiber types using the perturbative coupled-mode theory, which assumes that the mode fields are not deformed by the weak perturbation[60]. Here we present an illustrative example including the first six spatial modes. We show in Supplementary Fig. 10 the 55-mode case. Figure 3c–f, h–j displays the simulation results for the SI and GRIN MMFs, showcasing the linear interactions between the first six spatial modes (Fig. 3c, h) in curved fibers (Fig. 3d, i). The resulting linear coupling coefficient is presented in Fig. 3e, j. The phase mismatch is reflected by the effective refractive index $n_{eff}$ in Fig. 3f. Here we chose 1 cm as the bend radius in the simulation since it is the minimum value the fiber shaper can introduce by design; results including three different bending radii are displayed in Supplementary Fig. 10. These simulation results show that SI MMF has a greater $Q_{np}$ and smaller phase mismatch, both of which contribute positively to the strength of linear coupling (see Supplementary Fig. 10). This indicates richer spatiotemporal dynamics can be introduced in SI MMFs with the application of macro-bending. The significantly smaller $Q_{np}$ exhibited by GRIN MMFs can be attributed to the more confined mode field which sees weaker local perturbation at the center region (Fig. 3c, d, h, i) and hence higher resistance to mechanical deformation such as bending[61–64]. Moreover, the clustering nature of the propagation constants in GRIN MMFs makes the perturbation-induced mode coupling tend to occur between the nearly degenerate modes, commonly referred to as the degenerate-mode group[65]. Both factors make the multimodal nonlinear dynamics in GRIN MMFs less sensitive to fiber bending compared to that in SI MMFs.

Apart from the tunability, we also observed that the GRIN MMFs exhibit a lower threshold for laser-induced damage. Recent work has studied the effects of damage and transmission loss induced by multiphoton absorption on GRIN and SI MMFs[66,67]. As we gradually increased the input pulse energy for the GRIN fiber, we observed a decrease in total output power and a reduction in the output spectral

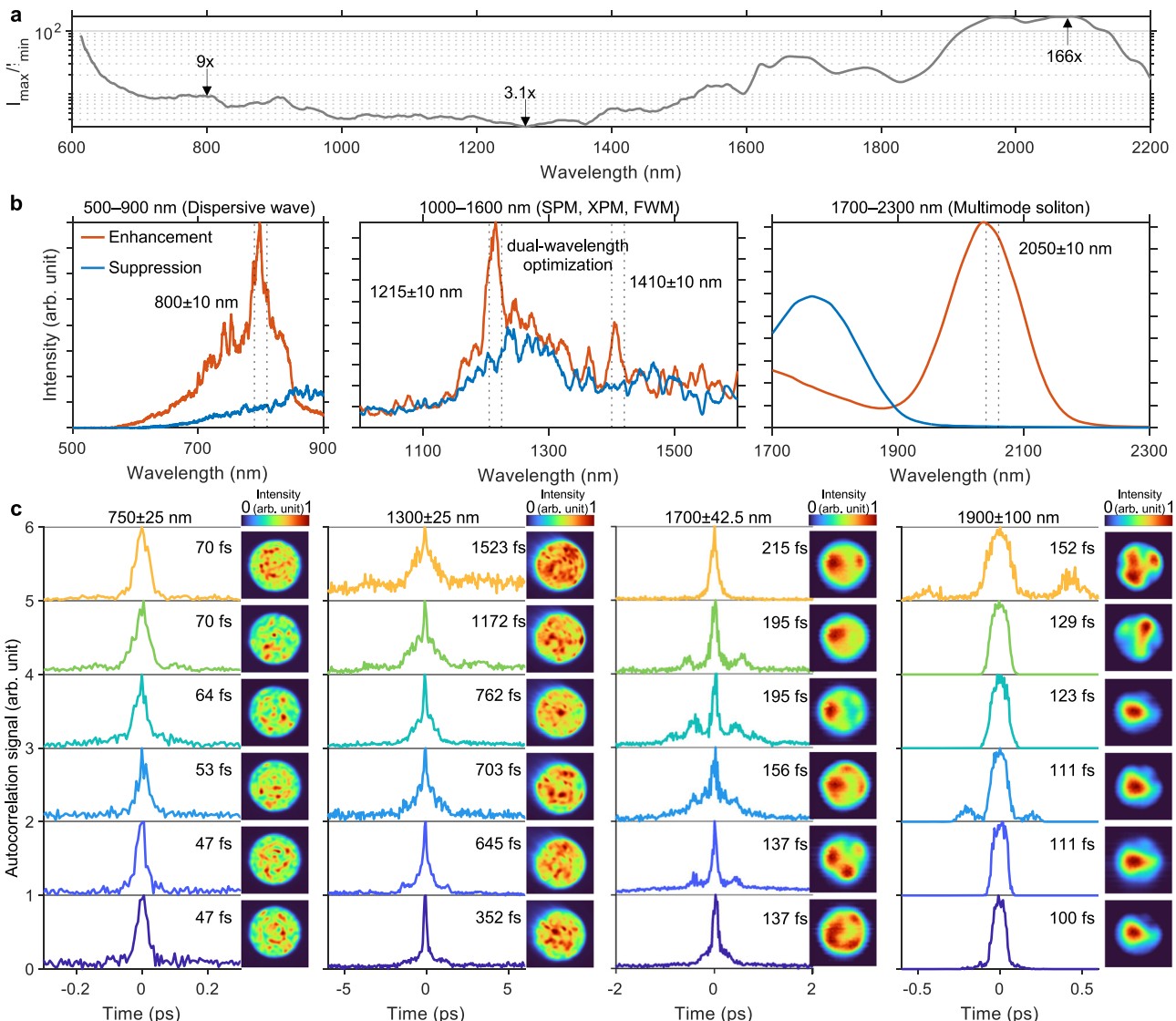

**Fig. 4 | Performance of fiber shaper: customization in spectral, temporal, and spatial domains. a** Ultrabroadband spectral tunability, evaluated from a set of spectra acquired from 3125 states of the fiber shaper. Representative examples of modulations in spectral band energy (**b**) of selected single bands and dual-band (denoted by vertical dotted lines), showing the most enhanced (Enhancement) and suppressed (Suppression) cases; and in pulse duration and the associated spatial intensity profiles (**c**) for selected spectral bands, with the center wavelength and bandwidth denoted above each panel, and the FWHM of the autocorrelation signals annotated near each trace. For visualization purposes, each intensity profile is normalized to its own maximum value. Source data are provided as a Source Data file.

span, suggesting the occurrence of fiber damage. This lower damage threshold is possibly due to the parabolic index profile which leads to more severe self-focusing effects[68], and the Germanium dopant in the GRIN fiber core[69]. These analyses and observations provide insights into the mechanism of fibre-shaper-based spatiotemporal control and entertain the choice of SI MMFs for a broadband high-peak-power tunable fiber source. Our observation, although unexpected, is consistent with the recent studies that show GRIN MMFs are not as sensitive to launching conditions or bending as SI MMFs[52,62–64]. Despite it being a drawback to applications that demand insensitivity to bending, the usually undesirable susceptibility to bending of SI MMFs, together with their power scalability, are essential for achieving effective spatiotemporal control of high-power pulse propagation in multimode fibers.

### Enhanced tunability: customizing the fiber output in spectral, temporal, and spatial domains

We next evaluate the effectiveness of using fiber shaper to control nonlinear effects in SI MMFs across different dispersion regimes. This

was done by examining, in distinct spectral regions, the tunability in spectral band energy (Fig. 4a, b), temporal duration (Fig. 4c), and spatial intensity profiles (Fig. 4c) of the output light field while shaping the fiber in real-time. These investigations were carried out under the same launching conditions (800 nJ, 46 fs, and 1300 nm) using the characterization apparatus depicted in Fig. 2. Figure 4a shows the spectral tunability factor across a continuous two-octave bandwidth, by exhausting the combination of actuator positions on the fiber shaper. The spectral tunability factor is defined as the ratio of the maximally enhanced intensity to the maximally suppressed intensity for each 20-nm-wide spectral band. We observed that the tunability varies substantially across the spectrum, with the minimum enhancement ratio of 3.1-fold shown in the middle of the spectral span, and significantly higher enhancement ratios ranging from 9-fold to as much as 166-fold at the two ends of the spectrum (below 800 nm and above 1500 nm). Such wavelength dependence is very likely to be the product of distinct broadening mechanisms in different dispersion regimes. To look into the versatility and effectiveness of the fiber shaper in manipulating various nonlinear effects, we next show in

Fig. 4b, c representative examples in different spectral regions, each featuring distinct dominant broadening mechanisms (see Supplementary Fig. 1a). The highest spectral tunability lies in the dispersive wave regime (500–900 nm; Fig. 4b) and the soliton regime (1700–2300 nm; Fig. 4b), the 9-fold and 166-fold marked in Fig. 4a are shown as instances in Fig. 4b. A recent study observed notable differences in the spectra for multimode solitons with identical energy but different initial mode components[70], which could provide insights into the high spectral tunability within the soliton regime. Simultaneous dual-band spectral tuning, with an averaged enhancement ratio of 2.3-fold for the two bands, is exemplified for the nonlinear phase modulation regime (1000–1600 nm; Fig. 4b).

Multimode solitons and their phase-matched dispersive waves consist of multiple mode combinations, within each combination the modes are group-velocity matched[49–52]. Mechanical perturbations such as a series of macro-bending can cause major changes in the few-mode composition (see Supplementary Fig. 15) and thus significantly alter spectral profiles in order to re-match the group velocity via mode-dependent spectral shifting. This is also manifested by dramatic changes of the few-mode-like spatial intensity profiles of multimode solitons (1700 ± 42.5 nm and 1900 ± 100 nm; Fig. 4c) and dispersive waves (750 ± 25 nm; Fig. 4c) across seven random states of the fiber shaper. These changes reflect the capability of the fiber shaper to significantly alter the mode composition, and thus the output light field in the spectral and spatial domains. The temporal duration at the fiber output, on the other hand, shows limited tunability as solitons and dispersive waves are intrinsically nearly transform-limited pulses. In contrast, the regime dominated by SPM, XPM, and FWM is known for highly nonlinear, strongly coupled, and highly multimodal behavior. The resulting spectrum is the incoherent summation of many modes, which makes spectral shape less sensitive to modal distribution changes and reduces the effectiveness of tuning the spectral intensity. The speckled spatial intensity profile (1300 ± 25 nm; Fig. 4c) corroborates the highly multimode nature of the nonlinear phase modulation process, in which the many-mode composition is needed to satisfy the intermodal phase-matching and velocity-matching conditions. Interestingly, the control in the temporal domain is much more significant in the nonlinear phase modulation regime than other regimes, ranging from 1523 to 352 fs based on the full-width at half-maximum (FWHM) of the autocorrelation signals, from 1080 to 250 fs assuming a Gaussian pulse shape for simplicity. This observation can be attributed to the effective reduction of modal dispersion in the multimodal output through optimization of fiber shaper-induced linear mode coupling.

Our results demonstrate that the fiber shaper effectively manipulates the modal compositions of the propagating pulse and the resulting spectral-temporal-spatial properties of the output light field. The intricate nonlinear interactions emphasize the benefits of using a fiber shaper to control the spatiotemporal dimension of the nonlinear dynamics in order to achieve a targeted output in different dispersion regimes. Earlier studies have successfully demonstrated broadband supercontinuum generation using silica MMFs in longer pulse duration regimes[8–11,21,71]. In this work, we demonstrate a two-octave tunable source in the femtosecond regime, which enables applications in multiphoton microscopy due to its high-peak-power pulses.

### Ultrabroadband high-peak-power tunable source for multiphoton microscopy

The results above demonstrate the potential of the fiber-shaper-controlled SI MMF for achieving an ultrabroadband high-peak-power tunable source, which is capable of modulating the output field in the spectral, temporal, and spatial domains. This capacity can facilitate diverse applications in spectroscopy, sensing, and imaging applications. As a proof-of-concept demonstration of such potential, we directly applied the proposed source for multiphoton microscopy

(MPM) (see Methods and Supplementary Fig. 11) to investigate whether it can be adaptively optimized for nonlinear imaging which demands sources with high spectral brightness (5–50 nJ)[72,73], short temporal duration (10–500 fs)[74,75], and confined spatial profiles[76].

We characterized the performance of the proposed source with fluorescent beads for two-photon fluorescence (2PF) and three-photon fluorescence (3PF) imaging. For comparison, we acquired reference images with a commercial laser source based on an optical parametric amplifier (OPA) from Light Conversion Cronus-3P. The results are presented in Fig. 5a–h, and the experimental setup which includes the optical path for imaging and the characterization of spectral, temporal, and spatial profiles of the fiber source, is depicted in Fig. 2. The raw output of the MMF was directed to a scanning MPM system after passing through selected bandpass filters (750 ± 25 nm for 2PF, 1300 ± 25 nm and 1225 ± 25 nm for 3PF). This setup can be readily improved in future studies with advanced wavelength selection mechanisms for more systematic and automatic multiband imaging. We observed that using the initial state of the fiber shaper, i.e., an unoptimized form, resulted in poor image quality (e.g. low multiphoton signals), which can be largely attributed to insufficient peak power due to the thinning of the energy distribution that comes with spectral broadening and/or temporal broadening caused by modal and chromatic dispersion.

By adaptively and coarsely optimizing the fiber shaper using feedback from the multiphoton signals, i.e. greedy search in our experiment, enhancements of 15-fold, 11-fold, and 6-fold in signals were achieved for 2PF (750 nm), 3PF (1300 nm), and 3PF (1225 nm) microscopy, respectively (Fig. 5c, f, h). To investigate the mechanisms of the improvement, we looked into the spectral, temporal, and spatial characteristics of the output pulses before and after optimization (Fig. 5j–r), and calculated the multiphoton signal generation efficiency according to the method described in[77] (Fig. 5s). To highlight the role of spectral and temporal tuning and for simplicity, the high-order dispersions and multimodal spatial compositions are neglected in the calculation and the pulse shape and spatial profile of the fiber source are assumed to be the same as those of the laser (see Methods and Supplementary Note 3). The strong agreement between the measured and the calculated improvements in signals confirms that pulse energy and pulse duration are the dominant factors affecting multiphoton signal generation efficiency in our experiments. Notably, the measured and calculated signal improvements for 2PF are more closely matched than those for 3PF. This can be attributed to the fact that the generation efficiency of 2PF is less dependent on the spatial distribution of the beam compared to 3PF[77] (see Supplementary Fig. 13).

Moreover, we observed that the output field optimization for signal enhancement of 2PF and 3PF varies as a result of different underlying mechanisms. Specifically, the signal enhancement for 2PF was primarily driven by a 3.8-fold increase in the spectral intensity at 750 nm, whereas the enhancement for 3PF signals was primarily due to a 2.7-fold (at 1300 nm) and 2.1-fold (at 1225 nm) reduction in pulse duration, resulting in peak powers of 0.72, 0.12, and 0.15 MW, assuming Gaussian pulse shape. This difference is consistent with our findings presented in Fig. 4, where the tunability in spectral and temporal domains exhibits strong spectral dependence due to the distinct dominant nonlinear effects: higher spectral tunability was observed in the dispersive regime (750 nm), whereas better temporal tunability was found in the nonlinear phase modulation regime (1300 and 1225 nm). Additionally, we observed that the corresponding spatial profile of 3PF showed a tendency to be less speckled and more confined when the pulse was shortened, due to a reduced modal dispersion. The reduction in speckling in 3PF is likely to result in less image blurring compared to 2PF (see Supplementary Fig. 12).

We next examined the performance of the proposed source for label-free imaging of freshly excised mouse tissue. We selected the 1225 ± 25 nm excitation band, which is not covered by most

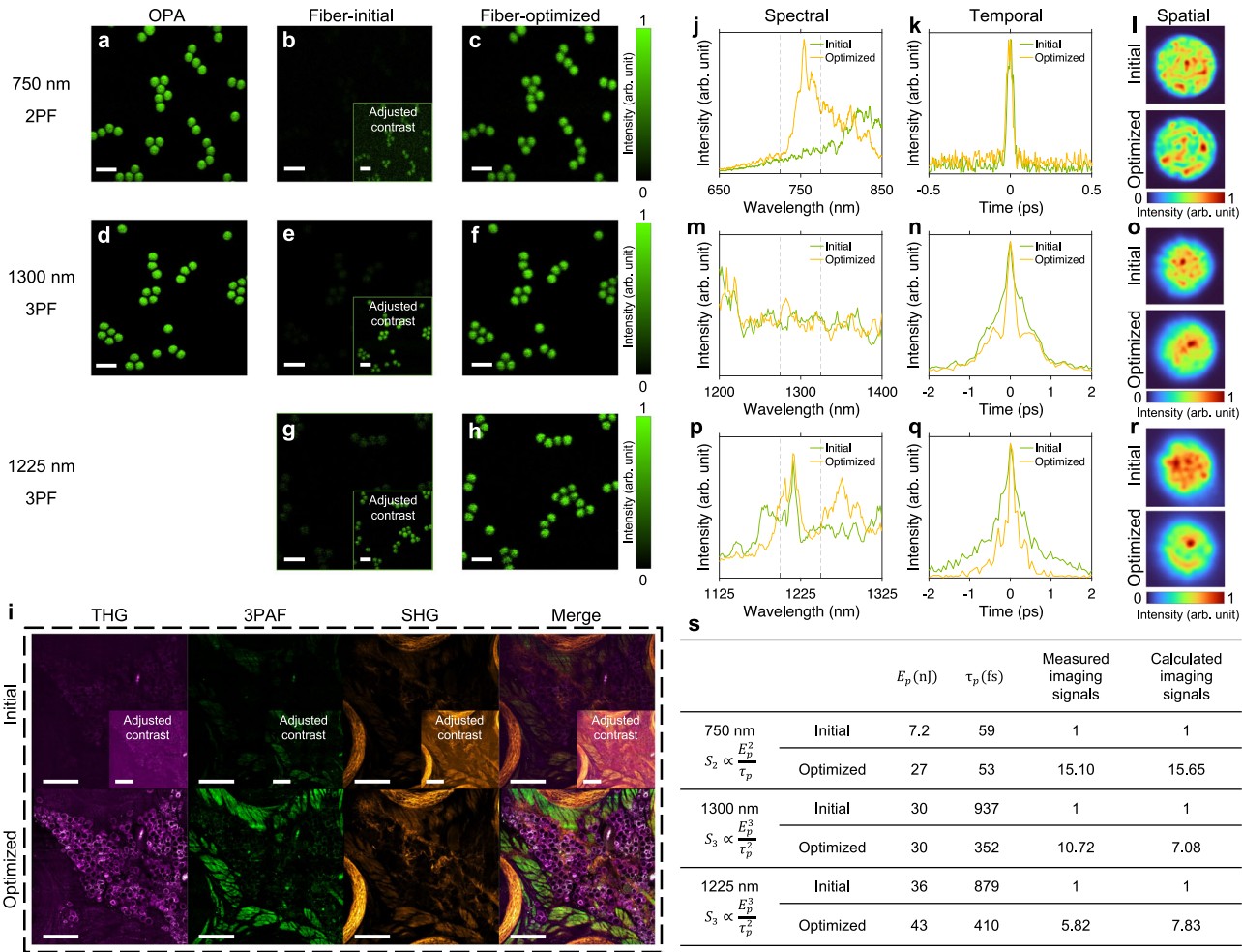

**Fig. 5 | Multiphoton microscopy with the fiber-shaper-controlled MMF source.** Demonstration of 2PF and 3PF imaging on fluorescent beads using OPA (**a**, **d**), initial fiber source (**b**, **e**, **g**), and optimized fiber source (**c**, **f**, **h**). The contrasts of the insets are adjusted for visibility. Scale bars: 20 μm. **i** Pseudo-color presentation of label-free imaging of fixed tissue from the mouse whisker pad at 1225 nm excitation using the initial and optimized fiber source, showing THG (magenta) signals of adipocytes, 3PAF (green) signals of muscles, and SHG (yellow) signals of collagen fibers. The contrasts of the insets are adjusted for visibility. Scale bars: 200 μm.

Characterization of the spectral (**j**, **m**, **p**), temporal (**k**, **n**, **q**), and spatial (**l**, **o**, **r**) properties of the output beam of the initial and optimized fiber source corresponding to the images in **a**–**h**. The 50-nm spectral band energy and the FWHM of autocorrelation signals are summarized in **s**; the spatial profiles are individually normalized to their maximum intensity. **s** Validation of the signal improvement from the images compared to the theoretical values calculated from pulse measurement. Source data are provided as a Source Data file.

commercial high-power OPA or low-rep high-peak-power lasers. After the same procedure of optimization (greedy search based on the imaging signals), the optimized SI MMF allows the simultaneous visualization of adipocytes through third-harmonic generation (THG), muscles through three-photon autofluorescence (3PAF), and collagen fibers through second-harmonic generation (SHG) (Fig. 5i), which demonstrated the potential of this source as a widely and continuously tunable source for bioimaging.

### Extension to other wavelengths

We have demonstrated above the effectiveness of using fiber shaper for spatiotemporal control of various nonlinear effects across different spectral regions, which leads to a high-peak-power source spanning the whole broadened spectrum. The broadened spectral span, however, exhibits strong wavelength dependence. For example, our results (see Supplementary Fig. 7) show that pumping in the normal dispersion regime (e.g., 800 nm) leads to limited spectral broadening and smaller spectral tunability, while pumping deeper into the anomalous dispersion regime (e.g., 1550 nm) results in a broad but discontinuous spectrum. To obtain an efficient and wide continuum for sufficient broadband spectral brightness, one often needs to pump the

waveguide in the anomalous dispersion regime close to its zero-dispersion wavelength (ZDW). For fused silica, the ZDW is close to 1280 nm. Although the main results of this study were generated using a pump wavelength of 1300 nm, the fiber shaper can be easily extended to other wavelengths - with comparable broadening and tunability - by pumping a multimode waveguide with ZDW close to the pump wavelength.

Here, to demonstrate the extension of the fiber shaper to other more accessible wavelengths such as 1040 nm, we leverage the blue-shifted ZDWs of the higher-order fiber modes to match the pump wavelength (see Supplementary Fig. 8). The results are presented in Fig. 6. The upper panel shows the spectral broadening with matched ZDW by exciting the $LP_{07}$ mode using a spatial light modulator (SLM)[78], resulting in a spectrum spanning from 550 to 1650 nm covering 1.6 optical octaves, which is comparable to the 2-octave spectral span we obtained in the previous sections. Inset shows the spectral tunability between 900–1700 nm, exhibiting great capability for spectral energy reallocation. In comparison, the reference spectrum obtained by weakly focusing without matching the pump and zero-dispersion wavelengths is displayed in the lower panel, limited spectral broadening was observed. Other methods to shift the ZDW include

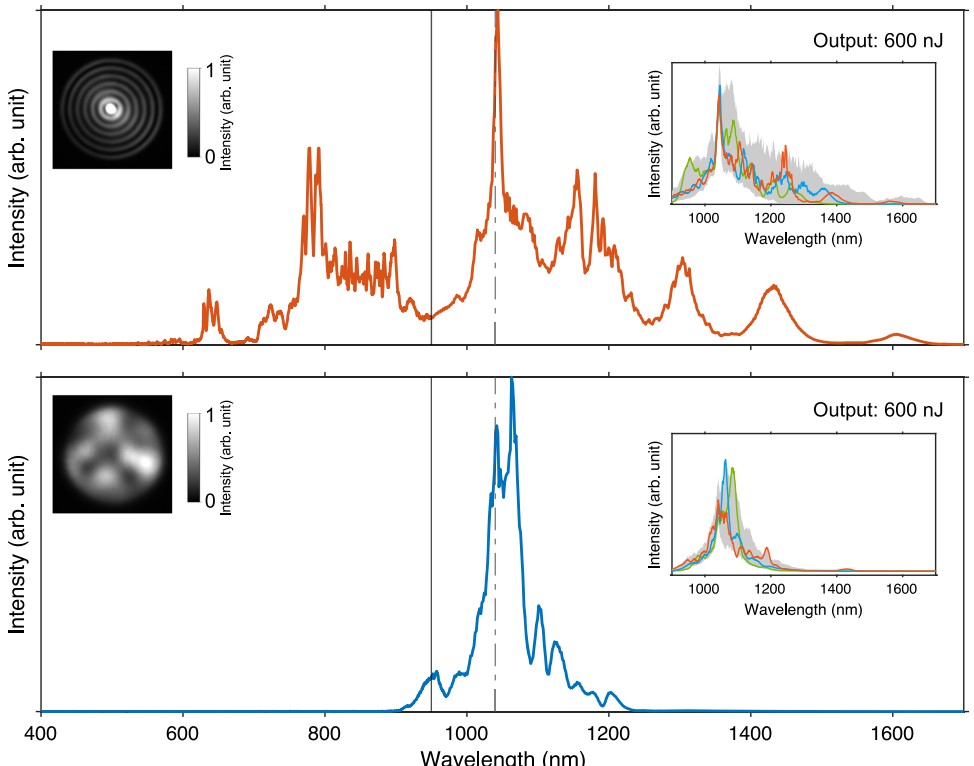

**Fig. 6 | Extension of fiber shaper to an input wavelength of 1040 nm.** 220-fs pulses were launched into the same standard silica SI MMF (50/125 µm, 0.22 NA) with a length of 30 cm. Normalized intensity patterns in gray scale represent the near-fields of the fiber output at low-power levels, indicating $LP_{07}$ and a mixture of lower-order modes were excited in the two cases, respectively. Output energy was fixed to 600 nJ. Inset shows the spectral tunability between 900 nm and 1700 nm, obtained with 1000 different fiber shape configurations. Vertical solid lines mark the spectral range measured by different spectrometers. The dash-dotted line denotes the input wavelength. Source data are provided as a Source Data file.

dispersion engineering and composing the waveguide with non-silica materials[21,79–81]. Through these techniques, more wavelengths can be potentially accessed with optimized spectral span and tunability.

## Discussion

In summary, we have presented a straightforward yet efficient way of controlling nonlinear effects by leveraging both the spatial and temporal degrees of freedom of the multimodal pulse propagation through a programmable fiber shaper. This method unlocks access to an even higher-dimensional space of spatiotemporal dynamics and two-octave-wide high peak power in an off-the-shelf SI MMF. The major contributions of this work include (1) opening up fresh perspectives for spatiotemporal control of nonlinear multimode pulse propagation, (2) proposing an avenue for power scaling and field control of broadband sources from visible to NIR-IR regime, and (3) providing accessible control designs that can be rapidly adopted for both nonlinear and linear modulation of multimode fibers.

Our proposed method directly modulates the modal interactions during the pulse temporal evolution using a slip-on 3D-printed fiber shaper, without additional non-fiber modulation layers such as free-space SLMs for wavefront shaping in spatial and temporal domains. The all-fiber route to single-stage nonlinear conversion inherently contributes to strong alignment robustness, great long-term stability (see Supplementary Fig. 2), and high spectral band energy. The fiber shaper excels in modulating the multimodal pulse propagation, which SLM cannot access easily. However, it is worth noting the SLM excels in modulating the input, which the fiber shaper cannot access easily (e.g., the broadening shown in Fig. 6). Because of the distinction of the modulation domains, these two approaches are complementary and can be implemented together to maximize control degrees of freedom.

Compared to the commonly used GRIN MMFs, the higher damage threshold and the significantly higher tunability of SI MMFs make it an essential part in spatiotemporal control of the high-power multimodal nonlinear effects. As a result, the proposed apparatus leads to a fiber source with great tunability in spectral (up to 166-fold reallocation), temporal (up to 4-fold shortening), and spatial domains. Furthermore, the fiber shaper enables combined spectral and temporal tuning, leading to high peak power levels across two-octave spectral bands. For instance, at 750 ± 25 nm, the peak power reached 0.72 MW and 0.15 MW at 1225 ± 25 nm, surpassing the expected regime of operation, which is usually the soliton regime for its inherently high spectral density and short pulse duration.

These performances (1) overcome the bandwidth limitation of existing high-peak-power fiber sources based on soliton formation[12,32,82–84] and (2) demonstrate orders of magnitude higher peak power compared to existing tunable fiber-based broadband sources[85]. These properties could benefit the emerging but technologically demanding applications in optical sensing, imaging, manipulation, and computing, that require light sources with broad spectral coverage, great tunability, ultrafast pulses (femtosecond-level pulse duration), stability, and/or high spectral density of energy and peak power.

## Methods
### Experimental setups and measurements
Pulses with an energy up to 800 nJ and duration of 40–60 fs were launched into 30-cm-long MMFs that were mounted on a three-axis translation stage, with the slip-on fiber shaper placed closely thereafter. This length of the fiber allows for effective spectral tuning without significantly dispersing pulses due to an unnecessarily long fiber (see Supplementary Fig. 14). A 6-cm doublet was used to form a 16-µm ($1/e^2$ radius) focal spot on the input facet of the MMFs, resulting in a coupling

efficiency of 85%. Two types of MMFs of the same core sizes were tested, namely the silica-core SI MMF (Thorlabs FG050LGA) and GRIN MMF (Corning OM4). The pulses out of the MMFs were split twice to couple into three different grating spectrometers after passing individual optical diffusers. The diffusers were used to reduce the spatial variance of the spectra out of MMFs. Each of these spectrometers covers a specific wavelength range: 192–1020 nm (Thorlabs CCS200), 899–1702 nm (Ocean Insight NIRQuest+1.7), and 858–2573 nm (Ocean Optics NIR256-2.5). To combine the three spectra obtained from the individual spectrometers, we selected 950 nm and 1600 nm as the reference points for "stitching". For each spectral region separated by these reference points, we first subtracted the background noises measured when the optical signals were blocked, then multiplied a scaling factor to the spectral traces to align their overall heights with those of the neighboring region at the stitching point. The pulse duration was measured using an intensity autocorrelator (APE PulseScope) based on the second-harmonic generation detection. The near-field beam profiles were acquired by two cameras. A CMOS-based (Mako G-040B) was used for the visible and near-infrared regions, and a thermal imaging camera based on a silicon microbolometer (DataRay WinCamD-FIR2-16-HR) was for longer wavelengths above 1700 nm. The optical powers were measured using a thermal power sensor (Thorlabs S425C-L). The fiber output is collimated by a parabolic mirror of 25.4-mm focal length (Edmund Optics 36-586), and then expanded by a pair of 4-f relay systems to slightly overfill the objective's pupil size (15.12 mm). A dichroic mirror (Thorlabs DMLP650L) is placed before the objective lens, which separates the excitation signals from the emission signals in the detection channels.

## Simulations

In all simulations, we assumed a single linear polarization for simplification. The simulated spectra in Supplementary Fig. 1b were acquired using the GMMNLSE[59]:

$$\frac{\partial A_p}{\partial z} = \mathcal{D}\{A_p\}$$
$$+ i\frac{n_2\omega_0}{c}\left(1 + \frac{i}{\omega_0}\frac{\partial}{\partial t}\right)\sum_{l,m,n}\left\{(1-f_R)S^K_{plmn}A_lA_mA_n^* + f_RS^R_{plmn}A_l\left[h*\left(A_mA_n^*\right)\right]\right\} \quad (2)$$

implemented with a numerical solver[58] in MATLAB. In the equation above, $A_p$ is the abbreviation of $A_p(z,t)$ representing the temporal envelope of the spatial mode-$p$; $\mathcal{D}$ denotes the linear propagation including dispersion effects. $S^K_{plmn}$ and $S^R_{plmn}$ are the mode overlap factors responsible to the instantaneous Kerr effect and Raman effect, determined by the spatial overlap of the four spatial modes involved. We set the nonlinear index $n_2 = 2.3\times10^{-20}$ m²/W and the Raman contribution $f_R = 0.18$; we included the self-steepening effect and up to the fourth-order linear dispersion effects. The mode and dispersion parameters of the SI MMF were calculated based on its pure silica core and its NA of 0.22.

The linear coupling coefficient $Q_{np}$ in Eq. (1) and Fig. 3e, j was calculated as

$$Q_{np} = \frac{k_0}{2n_{\text{eff}}}\iint\varepsilon_0\left[n^2(x,y) - n_0^2(x,y)\right]F_n(x,y)F_p^*(x,y)dxdy, \quad (3)$$

where the scalar function $F(x,y)$ represents the transverse fiber mode profile (see Fig. 3c, h). The mode parameters of the GRIN MMF were calculated based on a standard telecommunication-grade parabolic GRIN fiber used in[58].

## Fiber shaper design

To generate precisely controlled macro-bending to the fiber, we designed a device called fiber shaper. This device was fabricated using 3D printing (Stratasys Fortus 380mc) with acrylonitrile styrene acrylate material. The fiber shaper features a rectangular base with five slots that can hold the translating units (referred to as actuators in Fig. 1a) and constrain their linear motion in the desired direction. Each translating unit is powered by its own stepper motor (ELEGOO 28BYJ-48) through a 3D-printed rack and pinion system, allowing for individual control over their linear motion through a microcontroller (ELEGOO Mega R3) that communicates with a computer. To mount the fiber onto each translating unit, it is passed through a 0.5-mm-wide gap created by two disks and secured in place with a square cap. This design minimizes tension on the fiber during bending, and two half-disks with securing caps at the entrance and exit of the device ensure optimal functionality. The radius of the disk, which determines the minimum bend radius the fiber shaper can apply, is designed to be 10 mm to minimize bending-induced transmission loss. Different device parameters were tested to optimize the spectral tunability of the fiber shaper (see Supplementary Note 2 and Supplementary Figs. 4–6). As a result, the rich combinations of actuator translation displacements create a vast array of fiber shape configurations, allowing for effective utilization of the spatial and temporal degrees of freedom in controlling nonlinear pulse propagation in SI MMFs with high light throughput.

## Multiphoton microscopy

The multiphoton microscopy images were acquired using a custom-built inverted scanning microscope. The microscope used a pair of galvanometer mirrors (ScannerMAX Saturn-5 Galvo and Saturn-9 Galvo) to scan the beam. The beam was then focused by a water immersion objective (Olympus XLPLN25XWMP2, 1.05 NA). The emitted photons were collected using a photomultiplier (Thorlabs PMT2101). During imaging, the fiber source was pumped by 800 nJ 46 fs pulses at 1300 nm and 1 MHz repetition rate. The repetition rate of the OPA (Light Conversion Cronus-3P) was also fixed to 1 MHz during imaging. Fluorescent carboxyl polystyrene (Bangs Laboratories Inc FCDG009) was used as the imaging sample for characterization, a proper emission filter (Edmund Optics 530 ± 22.5 nm) was chosen to match its fluorescence emission spectrum. Fixed tissue from the mouse whisker pad was used in label-free imaging, where emission filters (Semrock 609 ± 28.5 nm, Edmund Optics 530 ± 22.5 nm, Thorlabs 405 ± 5 nm) were chosen for collecting the SHG, 3PAF, and THG signals, respectively. Raw images were loaded onto FIJI (National Institutes of Health) to apply pseudo-color maps. The bead sample images with 750 nm, 1225 nm, and 1300 nm excitation were respectively acquired with pixel dwelling times of 6 μs, 6 μs, and 10 μs, all with 150-μm field of view. The tissue sample images were acquired with a 30-μs pixel dwelling time and a 900-μm field of view, and 4 × 4 images were stitched to represent the mosaicked image. The greedy search method was implemented with the imaging signals as feedback. Specifically, the displacement of each actuator on the fiber shaper was sequentially chosen with the imaging signals maximized.

To calculate the signals of the bead sample images, we selected a few random regions of interest (ROI), each of which contained a spherical bead. The signals from measurement were calculated as the sum of pixel values within the ROIs. The theoretical value of the signals was determined by the multiphoton fluorescence generation efficiency based solely on pulse energy and duration as

$$S_n = C_n\frac{E_p^n}{\tau_p^{n-1}}(n = 2,3), \quad (4)$$

where $S_n$ refers to the $n$-photon generation efficiency, $C_n$ is the corresponding constant coefficient, $E_p$ and $\tau_p$ refer to the pulse energy and duration, respectively (see Supplementary Note 3).

## Data availability

The data that support the findings of this study are available from the corresponding author upon request and through collaborative investigations. Source data are provided in this paper. Source data are provided with this paper.

## Code availability

The codes that support the findings of this study are available from the corresponding author upon request and through collaborative investigations.

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

## Acknowledgements

The work has been supported by MIT startup funds. H.C. acknowledges support from the MIT Kailath Fellowship. K.L. acknowledges support from the MIT Jacobs Presidential. We thank Peter So for sharing with us their lab space and lasers during lab renovation, James Fujimoto for loaning the optical spectrum analyzer, and Phillip Keathley for loaning the autocorrelator and thermal camera. We would like to express our sincere gratitude to them and their students Matthew Yeung, and Lu-Ting Chou for valuable insights on nonlinear optics. We also extend our appreciation to Kristina Monakhova from our lab for providing constructive comments on the manuscript that greatly improved its clarity and quality. Our appreciation also goes to Miaomiao Jin from McGovern Institute for helping provide the tissue samples for our imaging experiments.

## Author contributions

T.Q., H.C. and S.Y. conceived the idea of the project. S.Y. supervised the research and obtained the funding. T.Q., H.C. and K.L. built the optical

setup and performed the experiments and simulations. M.L., E.L. and F.W. provided and prepared the tissue samples. T.Q, H.C., K.L., L.Y. and S.Y. wrote the manuscript with the input from all authors.

## Competing interests

S.Y., T.Q., H.C. and K.L. are coinventors of a patent (Application number: 63/470,554) on the method and apparatus for spatiotemporal control of light propagating through a fiber. The other authors declare no competing interests.
