## [Peer Review File · Nature Communications]

REVIEWER COMMENTS

Reviewer #1 (Remarks to the Author):

This work presents the first analysis of controlling non-linear effects in multimode fiber by introducing macro-bends at discrete locations along the fiber. This is an exciting new platform (sometimes called a fiber piano) for controlling modal interactions in multimode fiber and this work shows that it is particularly well suited for manipulating non-linear dynamics. The authors present a thorough study of the spectral, temporal, and spatial properties in different wavelength regimes and include an example application for multi-photon microscopy. I found the presentation of the spatial mode profiles in different spectral regimes to be a very nice visualization to help understand the different non-linear mechanisms at play. In general, this platform could be used to reach arbitrary spectral bands and is more accessible than spatial light modulator approaches, while providing additional degrees of freedom. This work also provided a nice demonstration of the advantages of using step-index MMF compared to GRIN for this type of application. I have only a few comments:

(1) In the introduction, the authors briefly mention similar work using piezo actuators (“fiber pianos”). Can you expand on the distinction between these techniques? I appreciate that this work is the first application of this scheme to controlling non-linear interactions, but I’m less clear on the trade-offs between the Piezo approach and the motorized actuator stages used here. The authors mention loss in the piezo based fiber pianos, but Ref. 37 also used a 1 cm minimum bend radius and similar step-index fiber (NA=0.22, 50 μm) and reported significantly higher bend loss (50%). Is the difference that in the current work, the authors are primarily exciting low-order modes?

(2) In this work, the authors studied a system using only 5 actuators. Can you comment on how the spectral/temporal tunability is expected to scale with the number of actuators? Supplementary Note 1 partially addresses this, but the tunability simply increases with the number of actuators up to the 5 tested. Will you continue to see a benefit increasing beyond 5 actuators? Can you project how much additional tunability is possible by using more actuators?

(3) I’m curious how the level of spectral/temporal/spatial control achieved here compares to what could be accomplished using traditional wavefront shaping with an SLM? Presumably this approach provides more degrees of freedom (or at least the potential for it as more actuators are added)—is there a way to quantify the resulting improvement in spectral/temporal/spatial control?

Reviewer #2 (Remarks to the Author):

Report on the paper: "Spatiotemporal control of nonlinear effects in multimode fibers for two-octave high-peak power femtosecond tunable source"

Authors: T. Qiu et al.

In this manuscript, the Authors apply to a highly nonlinear pulse propagation regime, the method introduced by W. Xiong et al in ref.(6) for the spatiotemporal control of light transmission in a multimode optical fiber (MMF) with strong linear mode coupling. The method is based on introducing controllable amounts of bending-induced stress to the fiber via a series of clamps. This work demonstrates that in this

manner it is possible to efficiently manage the spectral properties of supercontinuum generation (SCG), which results from the Kerr and Raman nonlinearities in a step-index (SI) MMF. In their implementation, the Authors use a programmable fiber shaper, which introduces multi-point macro-bending through five individually controlled motorized actuators applied to the MMF.

In the manuscript, the Authors have reported their extensive studies of the wavelength tunability of the SCG, by examining its impact on output beams, temporal profiles, and spectral characteristics. The authors have conducted a detailed investigation, by varying the actuators of the fiber shaper and input field wavelength, and compared the results with those obtained using graded-index (GRIN) fibers.

Notably, the authors harness this fiber-shaper controlled tunable source for multiphoton microscopy, and

have demonstrated significant enhancement of the signal from two-photon and three-photon absorption-induced fluorescence through adaptive optimization.

This study is comprehensive in its scope, and the manuscript is well-written. However, I would recommend it for publication in Nature Communications if the authors can better focus on the following topics:

(1) A similar strategy has been employed by Berti et al. (ref.1 below) to study the interplay of disorder introduced by strong linear mode coupling and Kerr nonlinearity, as far as the thermalization of multimode field is concerned. This work should be cited; besides, it should be explicitly stated if, in their implementation, the fiber shaper introduces strong (that is, between non degenerate modes) or weak (between degenerate modes) linear mode coupling? The distinction may be not so relevant for a

SI MMF, where most modes are non-degenerate, but it is important for a GRIN fiber. Could it be that the spectral shaping is not as effective for a GRIN MMF precisely because the linear mode coupling is not strong enough to couple its non-degenerate modes?

(2) Does the fiber shaper device have the capability to achieve near-single-mode output control?

(3) In line 118, the Authors state that the fiber shaper incurs minimal overall loss. As this loss been only measured in the linear regime? I wonder if the fiber shaper could induce significant mode-dependent loss for high-power pulses, potentially acting as a saturable absorber for a spatiotemporal mode-locked lasers.

(4) Could the fiber shaper also control the polarization of the output light?

(5) Can the fiber shaper mimic the functionality of a fiber piano, serving as both tunable bandpass filter and

dual-band filter?

(6) The authors exclusively examine the 30 cm length of MMF. What prompted the selection of this specific

length, and does it represent an optimal length that maximizes tunability?

(7) The authors conducted 3125 different measurements by adjusting the fiber shaper, as demonstrated in

Fig. 1(b). The algorithm or the automated search methodology employed for these extensive tests should

be elucidated. Maybe if the code was open source, it could have more impact. Additionally, what was the average calculation duration to obtain a single optimal output?

(8) Are there specific predetermined actuator positions that result in optimal outputs for specific wavelengths, like maximizing the spectrum at 1800 nm or 2000 nm?

(9) Sections 2.2 and the initial part of Section 2.3 appear to consist of standard tests without the utilization

of the fiber shaper, relying on older theories and results. The conclusions drawn from these sections appear relatively weak and do not contribute significantly to the core message of the paper. It might be beneficial to condense these sections and transition more swiftly to the primary topic, which is the tunability of the fiber shaper.

(10) Figure 4(f) displays only 5 modes, yet the authors made use of the first 6 modes in their analysis.

(11) In line 238, regarding the damage threshold of the GRIN fiber, I recommend the authors consider referencing (ref.2), here this threshold has been recently studied.

(12) In reference to the mode-content-dependent soliton spectral shifting discussed in lines 285-287, I suggest that the authors include a reference to ref.3. Its Authors have demonstrated that, in the presence of multimode solitons with identical energy but different initial mode components, there are notable differences in the resulting soliton spectrum. This observation could provide valuable insights into the substantial increase in the fold number (166-fold) observed within the soliton regime.

(13) Line 418: In line 418, could you provide insights into how the authors determined that the input mode corresponds to LP07? Additionally, in the context of the high-power regime, could you elaborate on the mode distribution at various main spectral peaks?

(14) On page 16, line 319, the Authors should mitigate their claim “To the best of our knowledge this is the broadest continuous supercontinuum (560-2200 nm)...” by referencing to earlier literature where similar or even broader SCG was demonstrated using either GRIN or SI MMFs (refs.4-6)

(15) On page 2, line 46, when mentioning nonlinear frequency generation with MMF, ref.7 which demonstrated frequency conversion via geometric parametric instability should be cited.

(16) It is suggested to add a couple of additional important review papers on nonlinear effects in MMF (refs.8-9)

Typos:

line 13: “Multimode fibers (MMFs) have recently reemerged as an attractive avenue”

⇒ “Multimode fibers (MMFs) have recently reemerged as attractive avenues”

line 34: “have become a renewed focus of interest” ⇒ “have become renewed focuses of interest”

line 60: “ignite advances of cutting-edge applications,” ⇒ “ignite advances in cutting-edge applications,”

line 90: “in linear and quantum regime” ⇒ “in linear and quantum regimes”

line 99: “were launched in to” ⇒ “were launched into”

Fig. 3: “the spectrum and beam profile at each input pulse energy is normalized to their” ⇒ “the spectrum

and beam profile at each input pulse energy are normalized to their”

Line 163: “The discrepancy in distribution of power spectral density” ⇒ “The discrepancy in the distribution of power spectral density”

line 338: “mercial laser source based on optical parametric amplifier” ⇒ “mercial laser source based on an optical parametric amplifier”

line 354: “15-fold, 11-fold and, 6-fold” ⇒ “15-fold, 11-fold, and 6-fold”

Fig. 7: “were launched into the the same standard silica” ⇒ “were launched into the same standard silica”

Fig. 7: “Inset shows the spectsal tunability beween” ⇒ “Inset shows the spectral tunability between”

List of references

1. N. Berti, et al. Interplay of Thermalization and Strong Disorder: Wave Turbulence Theory, Numerical Simulations, and Experiments in Multimode Optical Fibers. Phys. Rev. Lett.129, 063901 (2022). <https://doi.org/10.1103/PhysRevLett.129.063901>
2. M. Ferraro, et al., "Multiphoton ionization of standard optical fibers," Photon. Res. 10, 1394-1400 (2022). <https://doi.org/10.1364/PRJ.451417>
3. Yifan Sun, et al., "Multimode soliton collisions in graded-index optical fibers," Opt. Express 30, 21710-21724 (2022). <https://doi.org/10.1364/OE.459447>
4. K. Krupa, et al., "Spatiotemporal characterization of supercontinuum extending from the visible to the mid-infrared in a multimode graded-index optical fiber," Opt. Lett. 41, 5785-5788 (2016) <https://doi.org/10.1364/ol.41.005785>
5. S. Perret, et al, “Supercontinuum generation by intermodal four-wave mixing in a step-index fewmode fibre,” APL Photonics 1 February 2019; 4 (2): 022905. <https://doi.org/10.1063/1.5045645>
6. Z. Eslami, et al. Two octave supercontinuum generation in a non-silica graded-index multimode fiber. Nat Commun 13, 2126 (2022). <https://doi.org/10.1038/s41467-022-29776-6>
7. K. Krupa et al., “Observation of geometric parametric instability induced by the periodic spatial self-imaging of multimode waves,” Phys. Rev. Lett. 116, 183901(2016). <https://doi.org/10.1103/physrevlett.116.183901>
8. K. Krupa, et al., “Multimode nonlinear fiber optics, a spatiotemporal avenue,” APL Photonics 1 November 2019; 4 (11): 110901. <https://doi.org/10.1063/1.5119434>
9. L. G. Wright, et al. Physics of highly multimode nonlinear optical systems. Nat. Phys. 18, 1018–

Reviewer #3 (Remarks to the Author):

The paper deals with the control of multimodal nonlinear effects at high peak power levels: by leveraging not only the spatial but also the temporal degrees of freedom in step index MMFs using a programmable fiber shaper. This method represents from the point of view of the authors the first method that enables simultaneous access to the spatial and temporal degrees of freedom of multimodal nonlinear pulse propagation, achieving high tunability and broadband high peak power. Its potential as a nonlinear imaging source is further demonstrated by applying the MMF source to multiphoton microscopy, where widely tunable two photon and three-photon imaging is achieved with adaptive optimization.

Spatio-temporal analyses of propagation in multimode fibers have already been published and should be cited. As an example, the paper by K. Krupa et al. "Observation of geometric parametric instability induced by the periodic spatial self-imaging of multimode waves," *Phys. Rev. Lett.* 116, 183901 (2016); should be included as an example of a new nonlinear conversion process observed in optical multimode fiber which can drastically change the supercontinuum generation process.

In addition, another paper on temporal shortening in multimode fibers also published by Krupa et al. "Spatiotemporal light-beam compression from nonlinear mode coupling," *Phys. Rev. A* 97, 043836 (2018)" should also be cited as an early example of temporal versus spatial modal evolution.

The paper by Y Leventoux et al. "3D time-domain beam mapping for studying nonlinear dynamics in multimode optical fibers", *Optics Letters* 46 (1), 66-69 (2021), demonstrates the complex spatiotemporal interweaving of temporal and spatial effects in multimode fibers with nonlinear conversions.

Other papers on supercontinuum in multimode fibres have also been widely published and some references should be cited:

- Shalaby, B. M et al. "Visible Supercontinuum Generation by Dual-Wavelength Pumping in Multimode Rectangular Optical Fibers", *CLEO: Science and Innovations*, Paper# STh3N.4 (2014)
- Zhou, Renlai et al. "Half mJ Supercontinuum Generation in a Telecommunication Multimode Fiber by a Q-switched Tm, Ho:YVO4 Laser," *Journal of the Optical Society of Korea* 19(1) 7-12 (2015)
- Lopez-Galmiche, G. et al. " Visible supercontinuum generation in a graded index multimode fiber pumped at 1064 nm," *Optics Letters* 41(11) 2553-2556 View: (2016)
- Krupa, K. et al. "Spatiotemporal characterization of supercontinuum extending from the visible to the mid-infrared in a multimode graded-index optical fiber," *Optics Letters* 41(24) 5785-5788 (2016)

- Eftekhari, M. A. et al. "Versatile supercontinuum generation in parabolic multimode optical fibers," Optics Express 25(8) 9078-9087 See: (2017)

In this paper, the authors report successfully controlling nonlinear effects by exploiting spatial and temporal degrees of freedom through programmable fiber shaping. This is clearly not the case from my point of view. They modified the curvature of the fiber, which affects the spatial and then temporal propagation of the short pulse. There is a big difference between modifying the spatio-temporal evolution and controlling the spatio-temporal profile. This must be said and explained correctly. As an example, there is no good control of the modal excitation, only a curvature of the fiber which creates a linear coupling between modes. This coupling is not controlled in order to select the number of modes or the modes desired for propagation.

Are the authors able with this system to excite one or two given modes with temporal control of the duration of the input pulse?

Are the authors able with this system to excite only high order modes with control of the pulse profile?

Are the authors able with this system to excite in output only low order modes by fixing the spectrum profile and a precise pulse duration?

In the time domain, there is no control of the coupled pulse in the fiber. No control over the shape or duration of the pulse is clearly explained. Only propagation modification using actuators is used. It is then evident that due to the additional bending and stress on the fiber, the energy distribution across the transverse modes will be changed, which directly changes the time domain due to the short pulse used in the experiment. Beyond a modification of the experimental propagation conditions, no control is really demonstrated. Can the authors show the modal content based on the actions of the actuators? For example, can authors impose pulse duration and spatial mode content on their actuators?

Authors claim line 68 that they introduce a new way of controlling multimodal nonlinear effects. How is it possible without controlling the modal content of the beam and the initial pulse duration and shape?

They claim also (line 66) that they realized, in high-power regimes by exploiting not only the spatial but also the temporal degrees of freedom a control of the output beam.

They claim also to control of the brilliance of the output beam by never showing a measure proving that evolution. What is the M^2 coefficient of the output beam versus activating actuators?

To me, this paper is just a demonstration that short pulse coupling in MMF can be modified along the propagation by manipulating the curvature at many propagation distances. Is it possible with this system to shorten or increase the pulse duration in relation to the modal content? Is it possible to precisely control modal content?

The authors state at line 111 that: "By producing local index perturbations, the actuators cause an energy coupling between modes (39, 40) at several points in the evolution of the pulse. This change does not prove a control of the energy exchanges between modes and a control of the characteristics of the pulses.

What means "at multiple time points"? Is it at different positions in fiber? If so this is not a temporal control along the pulse duration.

In Figure 1 the authors show different spectra depending on the actuations of the actuators. Is it possible to start from the spectral profile we need and manipulate the actuators to obtain the result?, with a given pulse shape and duration?

The authors, in line 130, stated that solitary waves consisting of multiple spatial modes begin to form..." Can the authors give a proof of the soliton contents? Is it possible to control this soliton content with the actuators?

What are the losses introduced by the actuators on the propagating output beam and on each modes. Can the authors give a coupling coefficient between transverses modes by using actuators?

The figure 3 show a standard supercontinuum obtained at the fiber output. What is the novelty here?

In Section 2.3, the authors claim that actuators modify the local refractive index profile, causing energy coupling between the modes at several points in time during the evolution of the pulse, which allows in total to high-dimensional spatio-temporal control of multimode nonlinear pulse multiplication"; Where is the control here. For example, is it possible to fix the coupling between modes LP01 and LP21 or other pairs of modes?

The authors claim again line 200 that the spatial and temporal degrees of freedom

of nonlinear pulse propagation can be simultaneously controlled. This is not true. They are changing the linear coupling between modes which affect the pulse shape, but no control is really shown here. As an example is it possible to shorten the pulse duration or to choose a given pulse duration with energy only on the LP01 mode?

The authors claim also line 238 that: “ Apart from the tunability, we also observed that the GRIN MMFs exhibit a lower threshold for laser-induced damage. As we gradually increased the input pulse energy for the GRIN fiber, we observed a decrease in total output power and a reduction in the output spectral span, suggesting the occurrence of fiber damage. This lower damage threshold is possibly due to the parabolic index profile which leads to more severe self-focusing effects (56), and the Germanium dopant in the GRIN fiber core (57).”

This statement is obvious and everyone says that the modal area of lower order modes is different in step and grin fibers.

The authors stated that: “The high peak power of broadband (approaching MW on average) was achieved by combining spectral energy reallocation (up to 166 times) and temporal shortening (up to 4 times). Such capabilities are only enabled by the fiber shaper, a single fully fiber modulator without the need for 3 external wavefronts.”

Just to show that this assertion is slight, it is enough to take the transformation of a normal pulse during the propagation in abnormal regime to show its transformation into solitons whose peak power can be very largely increased by the simple effect of the nonlinearity and dispersion. This statement is therefore true but very widely known without the authors putting this propagation regime into perspective of their experience.

Reviewer #4 (Remarks to the Author):

The authors propose an intriguing method for achieving flexible spatiotemporal modulation of high peak power femtosecond laser light pulses using step-index (SI) multimode fibers (MMFs) through nonlinear optical processes. Despite the simplicity of implementing this method by bending the SI MMFs with a

low-cost “fiber shaper”, it has achieved broad 2-octave spectral modulation at its maximum. The careful examination of the spatiotemporal modulation has demonstrated the advantages of this proposed method, including simplicity, broad spectral tunability, and high power tolerance, compared to previously reported methods using graded-index (GRIN) MMFs. While the section describing the profile of the modulated output has been thoroughly investigated, there is room for improvement in the section describing its application in multiphoton imaging.

I hope that the comments below will be useful for the authors.

Major comments

1. The Methods section lacks a clear explanation of the multiphoton imaging setup. The spatial intensity profile of the output beam (shown in Fig 6 l, o, r) does not exhibit typical distributions like Gaussian due to the influence of the MMFs and the modulations. Especially in that case, which spatial range of the beam is introduced to the objective pupil is important to evaluate the imaging quality because the spatial distribution of the beam influences on the PSF. I recommend the authors describe how large the beam diameter ($1/e^2$ radius) of the output laser light expanded by the relay optics (beam expander) inside the multiphoton microscopy and what proportion of the diameter of the output beam is introduced. Also, please provide information about the dichroic mirror and the parabolic mirror.

2. In Fig. 6s, fluorescent images are quantitatively evaluated, and their intensity is compared with theoretical values. While the 2PF result showed a good match, the 3-photon applications did not. This discrepancy may arise from the lack of consideration for differences in the point spread functions (PSFs). Personally, I would prefer to estimate the PSFs using fluorescent beads below the diffraction-limited size, which could provide insights into the imaging quality of 3-photon applications compared to the 2PF, as the authors mentioned in L385-386.

Minor comments

1. The y-axis label of Fig. 5c is misspelled as A”o”tocorrelation Signal.

2. Add a color code to the spatial intensity profiles in Fig. 5c, 6i, 7, and Supplementary Fig. 2.

3. Provide an explanation for the input wavelength in Supplementary Fig. 3-5.

4. Increase the contrast of the 3125 different states in the figures to highlight their broad tunability.

5. For “reader friendly”, at least, explain the variables A_n , β , and z in the equation (1) in the main text, even though they are explained in ref. 47.

Response to Review

We would like to thank the editor and the reviewers for finding time to go through our paper and for providing valuable comments and suggestions. We have added new experiments and analyses, and revised the manuscript and supplementary materials to address all the concerns and suggestions raised by the reviewers. The manuscript is now improved in clarity and completeness as a result of the feedback from the reviewers. The important corrections and additions have been highlighted in red in the revised manuscript to facilitate the re-review.

Reviewer #1 (Remarks to the Author):

This work presents the first analysis of controlling non-linear effects in multimode fiber by introducing macro-bends at discrete locations along the fiber. This is an exciting new platform (sometimes called a fiber piano) for controlling modal interactions in multimode fiber and this work shows that it is particularly well suited for manipulating non-linear dynamics. The authors present a thorough study of the spectral, temporal, and spatial properties in different wavelength regimes and include an example application for multi-photon microscopy. I found the presentation of the spatial mode profiles in different spectral regimes to be a very nice visualization to help understand the different non-linear mechanisms at play. In general, this platform could be used to reach arbitrary spectral bands and is more accessible than spatial light modulator approaches, while providing additional degrees of freedom. This work also provided a nice demonstration of the advantages of using step-index MMF compared to GRIN for this type of application. I have only a few comments:

Response: We appreciate the reviewer's positive and thoughtful comments.

(1) In the introduction, the authors briefly mention similar work using piezo actuators (“fiber pianos”). Can you expand on the distinction between these techniques? I appreciate that this work is the first application of this scheme to controlling nonlinear interactions, but I’m less clear on the trade-offs between the Piezo approach and the motorized actuator stages used here. The authors mention loss in the piezo based fiber pianos, but Ref. 37 also used a 1 cm minimum bend radius and similar step-index fiber (NA=0.22, 50 μm) and reported significantly higher bend loss (50%). Is the difference that in the current work, the authors are primarily exciting low-order modes?

Response: We thank the reviewer for appreciating the novelty of this work in proposing a new platform for controlling multimodal nonlinear effects in fibers. Regarding the difference in the reported loss induced by bending, there are a few potential factors:

1. **Difference in the bending design.** In our work, we opted for a bend radius that is designed to be equal to or larger than 1 cm to minimize loss coming from micro-bending, tension,

and stress, as reported in literature [1]. While a similar bend radius might have been used in Ref. 37 (estimated to be approximately 1cm under the three-point contact assumption), the small contact area between the fiber and the post in the piezo setup is more likely to induce micro-bending or stress than the large contact area between the fiber and the bending structure (the 3D-printed base) in the motorized setup, which might be advantageous in introducing stronger mode coupling effects but disadvantageous for maintaining low loss.

2. **Cumulative effect.** Such bending-related loss could be exacerbated with an increasing number of benders. In Ref. 37, a total of 28 piezo actuators were implemented in the fiber piano system. If every actuator introduces 2.4% loss (based on a numerical guess derived from the reported 50% total loss) due to micro-bending or stress, the fiber piano that is made of 28 piezo actuators would result in 50% of total loss.
3. **Exciting lower-order modes.** Although weakly focusing is used to launch the pulses into the MMF, higher-order modes can be observed in the output because of the disorder and bending-induced mode coupling. In addition to the measured output spatial profiles (Supplementary Fig. 3), to gain more insights into if and how lower-order modes could couple to higher-order modes using different fiber shaper configurations, we have performed simulations to demonstrate the effect of linear mode coupling in relation to the bending radius. As shown in Response Fig. 1, relatively strong mode coupling happens with the bending radius ranging from 10 cm to 1 cm, which results in increased higher-order modal content in the output despite the initial weakly focusing launching condition. Therefore, initial weakly focusing exciting condition is likely not a factor for low loss in this setup.

Response Figure 1 (Added as Supplementary Figure 10): Simulation results of bending-induced mode coupling in step-index (SI) and graded-index (GRIN) multimode fibers (MMFs). The first two columns show the effective refractive index n_{eff} and the coupling coefficients Q_{np} for the first 55 spatial modes of

the SI and GRIN MMFs. The right panel shows the linear mode coupling strength $|Q_{np}/\Delta\beta_{np}|$ resulting from macro-bending at various bending radii (1 cm, 10 cm, 100 cm). Modes in the GRIN MMF are clustered as the degenerate-mode group, indicated by the solid white lines.

The tradeoff between the piezo approach and the motorized actuator approach was briefly summarized in the first factor above. More detailed description is described below.

The main motivation of the fiber shaper design is to minimize optical loss while introducing sufficient mode coupling effects. First, in order to minimize loss, we intend to make the bending as smooth as possible and minimize the chance of micro-bending and mechanical stress during bending. This can be achieved by the design of a large-radius (1-cm contact radius) and flat base plate as the bending structure, which facilitates a larger contact area between the fiber and the base plate and thus enables smooth bending of fibers. In contrast, the small contact area between the fiber and the post (~1.3-mm contact radius, eyeballing from the paper figure and the physical size of the bender) in the piezo setup is more likely to induce micro-bending or stress, which might be advantageous in introducing stronger mode coupling effects but disadvantageous for maintaining low loss.

Secondly, the use of a larger contact radius (1 cm) necessitates the use of actuators with larger motion range (>1 cm) to create sufficient bending in order to introduce significant mode coupling effect. This motion range is challenging for piezoelectric systems, which features high precision but a constrained motion range, usually under 1 millimeter. In contrast, motorized actuator systems provide a considerably greater motion range (often over 1 centimeter), which is effective for creating sufficient mode coupling effects with low optical loss (in combination with the large contact radius) for our application. The last minor factor is that the piezo approach can be costly due to the need of high driving voltage, which makes the low-cost motorized actuator setup advantageous for high scalability and accessibility in the future.

Accordingly, we have revised our manuscript and figures as follows:

- Revised the description of the differences between the fiber piano and the fiber shaper in our setup in Introduction Section 1.
- Added Response Figure 1 as Supplementary Figure 10.

(2) In this work, the authors studied a system using only 5 actuators. Can you comment on how the spectral/temporal tunability is expected to scale with the number of actuators? Supplementary Note 1 partially addresses this, but the tunability simply increases with the number of actuators up to the 5 tested. Will you continue to see a benefit increasing beyond 5 actuators? Can you project how much additional tunability is possible by using more actuators?

Response: We appreciate the reviewer's comments regarding the scalability of spectral/temporal tunability with the number of actuators. The choice of the number of actuators is the outcome of

the tradeoff between the pulse duration preferences of multiphoton microscopy (MPM) and the design constraints of the fiber shaper system. On one hand, a shorter fiber is beneficial for maintaining a short temporal pulse duration in MPM for efficient multiphoton excitation [2]. As demonstrated in Response Fig. 2, when we increased the fiber length from 30 to 40 cm, a significant pulse broadening occurs, which is unfavorable for high-order nonlinear processes (e.g., three-photon absorption). Meanwhile, a short fiber poses a challenge to the system design due to the limited space for the fiber shaper, as each actuator takes a space of around 2 cm to implement in the current setup. We believe that more control degrees of freedom can be introduced by customizing miniaturized actuators to fit in the limited space.

Response Figure 2: Representative results of the comparison of pulse durations out of straight step-index multimode fibers with two different fiber lengths. The measurements were performed on the bandpass filtered (1100 ± 25 nm) output with 1040 nm pump as an example.

Increasing the number of actuators theoretically enhances the device's control degree of freedom. As shown in Response Fig. 3, the spectral tunability keeps increasing when we expand the number of actuators to eight, indicating potential for scalability. However, it is important to note that while the expansion of spectral tunability is approximately linear to the number of actuators, the number of solution states grows exponentially with the number of actuators. This results in a tradeoff between the control degrees of freedom (solution search space) and the output degrees of freedom (tunability). The ultimate choice can depend on the applications, for example, if a study prefers more tunability and has no need of short pulses, more actuators can be used. If a study prefers shorter pulses and a higher efficiency of converting control degrees of freedom to output degrees of freedom, fewer actuators can be used.

Response Figure 3 (Added as Supplementary Figure 9): Spectral tunability with respect to the number of actuators. (a) The relationship between spectral tunability and the number of actuators. The tunability score is quantified by the mean spectral tuning range across the entire wavelength spectrum $\frac{1}{N} \sum_i [(I_{\lambda_i}^{\max} - I_{\lambda_i}^{\min}) / \bar{I}_{\lambda_i}]$. (b) All spectral data and the relative intensity variation at each wavelength $(I_{\lambda_i} - \bar{I}_{\lambda_i}) / \bar{I}_{\lambda_i}$.

In summary, considering the balance between shorter pulse duration and tunability, we opted to construct the device with five actuators. This configuration allows for effective fiber output customization without significantly compromising the short pulse duration that is needed for higher-order multiphoton imaging. However, this can be further optimized. For example, if we can reduce the size of each actuator and fit more actuators within smaller fiber lengths, we could produce more tunability without compromising the output pulse duration.

We have revised our manuscript and figures as followed:

- Added discussions about the scalability of the tunability in Results Section 2.2.
- Included Response Figure 3 as Supplementary Figure 9.

(3) I'm curious how the level of spectral/temporal/spatial control achieved here compares to what could be accomplished using traditional wavefront shaping with an SLM? Presumably this approach provides more degrees of freedom (or at least the potential for it as more actuators are added)—is there a way to quantify the resulting improvement in spectral/temporal/spatial control?

Response: Yes, the reviewer is correct that, by tackling the multimodal pulse propagation, the fiber shaper is designed to access more degrees of freedom in the control space of nonlinear effects in

MMFs. Borrowing the concept from the fiber piano work in the linear regime [3], the fiber piano/shaper approach offers a higher degree of control as it directly modulates the transmission matrix of a fiber with N^2 degrees of freedom (where N is the number of eigenmodes), as opposed to the wavefront shaping approach which modulates the input with $2N$ degrees of freedom. This effect is even more dramatic in the nonlinear regime [4]. Another analogous example for our multi-point bending design is Multiple-Plane Light Conversion (MPLC) [5], which allows for more control degrees of freedom by perturbing/modulating the light propagation sequentially at different propagation distance. However, despite the theoretical potential, it remains challenging to have an exact and conclusive comparison in the total output degrees of freedom because both approaches (i.e., SLM or time-dependent disorder) are yet far from fully developed and both have significant room for grow in terms of hardware and algorithm design. Our observation, based on what we have learned so far, is that SLM excels in modulating the input, which the fiber shaper cannot access easily, whereas fiber shaper excels in modulating the multimodal pulse propagation, which SLM cannot access easily. Because of the distinction of the modulation domains, these two approaches are complementary and can be implemented together to maximize control degrees of freedom. We are actively exploring the integration of both technologies in a single system and will provide a more comprehensive analysis in future work.

We have added discussions about the differences and complementary properties of SLM and the fiber shaper in Discussion Section 3.

Reviewer #2 (Remarks to the Author):

In this manuscript, the Authors apply to a highly nonlinear pulse propagation regime, the method introduced by W. Xiong et al in ref.(6) for the spatiotemporal control of light transmission in a multimode optical fiber (MMF) with strong linear mode coupling. The method is based on introducing controllable amounts of bending-induced stress to the fiber via a series of clamps. This work demonstrates that in this manner it is possible to efficiently manage the spectral properties of supercontinuum generation (SCG), which results from the Kerr and Raman nonlinearities in a step-index (SI) MMF. In their implementation, the Authors use a programmable fiber shaper, which introduces multi-point macro-bending through five individually controlled motorized actuators applied to the MMF.

In the manuscript, the Authors have reported their extensive studies of the wavelength tunability of the SCG, by examining its impact on output beams, temporal profiles, and spectral characteristics. The authors have conducted a detailed investigation, by varying the actuators of the fiber shaper and input field wavelength, and compared the results with those obtained using graded-index (GRIN) fibers. Notably, the authors harness this fiber-shaper controlled tunable source for multiphoton microscopy, and have demonstrated significant enhancement of the signal

from two-photon and three-photon absorption- induced fluorescence through adaptive optimization.

This study is comprehensive in its scope, and the manuscript is well-written. However, I would recommend it for publication in Nature Communications if the authors can better focus on the following topics:

Response: We thank the reviewer for the positive feedback and appreciate the reviewer's constructive comments and suggestions for improvement. We have provided our point-by-point responses below.

(1) A similar strategy has been employed by Berti et al. (ref.1 below) to study the interplay of disorder introduced by strong linear mode coupling and Kerr nonlinearity, as far as the thermalization of multimode field is concerned. This work should be cited; besides, it should be explicitly stated if, in their implementation, the fiber shaper introduces strong (that is, between non degenerate modes) or weak (between degenerate modes) linear mode coupling? The distinction may be not so relevant for a SI MMF, where most modes are non-degenerate, but it is important for a GRIN fiber. Could it be that the spectral shaping is not as effective for a GRIN MMF precisely because the linear mode coupling is not strong enough to couple its non-degenerate modes?

Response: We have added ref.1 to the main text in Introduction Section as reference 26.

The reviewer is correct that mode coupling effects from the current bending device is not strong enough to couple significantly to the non-degenerate modes in GRIN fibers even though it is effective in doing so in SI fibers. This is because the strength of mode coupling effects collectively depends on the amount of the applied perturbation (i.e., fiber bend radius in this case) and the spatial modal profiles and the distribution of the modal propagation constants of the MMFs.

To further illustrate this point, we performed numerical simulations to examine the linear mode coupling strength resulting from macro-bending of SI MMFs and GRIN MMFs at various bending radii, including 1 cm, 10 cm, and 100 cm. The coupling strength is defined as $|Q_{np}/\Delta\beta_{np}|$ [6], and is simulated under the perturbative coupled-mode approximation. As shown in Response Fig. 1, the SI MMF exhibits stronger mode coupling effects than the GRIN MMF under similar bending conditions. This discrepancy can be attributed to the smaller spatial mode profiles and the larger spacing of modal propagation constants associated with GRIN MMFs (Fig. 3). Therefore, given the same bending radius, mode coupling effects in SI MMFs is more significant than that in GRIN MMFs. Also, as expected, the strength of mode coupling also depends on the bending radii, as the higher curvature leads to more significant perturbations in the local refractive index. We can expect more significant mode coupling effects to show up in GRIN MMFs if a smaller bend radius is employed.

Response Figure 1 (Added as Supplementary Figure 10): Simulation results of bending-induced mode coupling in step-index (SI) and graded-index (GRIN) multimode fibers (MMFs). The first two columns show the effective refractive index n_{eff} and the coupling coefficients Q_{np} for the first 55 spatial modes of the SI and GRIN MMFs. The right panel shows the linear mode coupling strength $|Q_{np}/\Delta\beta_{np}|$ resulting from macro-bending at various bending radii (1 cm, 10 cm, 100 cm). Modes in the GRIN MMF are clustered as the degenerate-mode group, indicated by the solid white lines.

Accordingly, we have revised our manuscript and figures as followed:

- Added ref.1 to the main text in Introduction Section as reference 26.
- Added Response Figure 1 as Supplementary Figure 10.

(2) Does the fiber shaper device have the capability to achieve near-single-mode output control?

Response: In our experiments, particularly in the soliton regime, we observed that the spatial profile showed nearly single LP_{01} mode after optimization of the shaper, as shown in Response Fig. 4. However, it is important to note that in this soliton regime, the beam profiles are initially few-mode. If the target band is in the nonlinear phase modulation regime, where the output beam profiles are highly multi-mode, it would be more challenging to achieve near-single-mode output control with our current fiber shaper setup.

Response Figure 4: Near-single-mode output in the soliton regime after optimization of the shaper.

(3) In line 118, the Authors state that the fiber shaper incurs minimal overall loss. As this loss been only measured in the linear regime? I wonder if the fiber shaper could induce significant mode-dependent loss for high-power pulses, potentially acting as a saturable absorber for a spatiotemporal mode-locked lasers.

Response: We are grateful for the reviewer's insightful suggestions regarding the potential as a saturable absorber. We are enthusiastic about the idea.

Regarding the measurement of loss, we would like to clarify that our loss measurements were conducted in the high-power nonlinear regime, under pump conditions of 1300 nm wavelength, 800 nJ energy, and 46 fs duration. Even in this high-power regime, we found that the overall loss incurred by the fiber shaper remained negligible. Our primary objective in designing the fiber shaper was to deliver high-peak-power pulses to our multiphoton imaging system. Consequently, we designed the fiber shaper parameters to minimize bending loss while maximizing tunability (see more in the Response to Reviewer 1 Comment 1).

Although we did not observe significant mode-dependent loss, we agree with the importance of investigating the potential of tuning the overall modal content using fiber shaper. As a proof-of-concept demonstration, we employed a few-mode fiber to illustrate the potential for tuning modal content. Regarding the mode-dependent loss due to leakage from the fiber core, it is worth noting that it has been demonstrated that bending can indeed induce higher loss to higher-order modes in multimode fibers [1].

Herein, we presented a proof-of-concept demonstration using a few-mode fiber (Thorlabs FG010LDA) in Response Fig. 5. With a mixture of three linearly polarized (LP) modes (LP_{01} , LP_{11a} , LP_{11b}) as the initial excitation beam, the output spatial profiles of a 10- μ m core step-index MMF (SI10) showed distinct spatial profiles under different bending conditions. It was attributable to the combined effects of mode-dependent loss and mode coupling. Notably, under certain bending states, we observed near-single-mode output of LP_{11a} (Response Fig. 5b), LP_{11b} (Response Fig. 5d), and LP_{01} (Response Fig. 5g). Such mode selectivity shows the promise of turning MMFs into a saturable absorber with the proposed fiber shaper. To further the development, more in-depth studies are needed to provide comprehensive understanding, as the result demonstrated in Response Fig. 5 is the outcome of the combinatory effects of mode-dependent loss and mode coupling.

Response Figure 5: Normalized output spatial profiles of a 10- μm core step-index multimode fiber under various bending conditions. Each profile is individually normalized. The initial output beam profile shows a mixture of approximately three linearly polarized (LP) modes (LP_{01} , LP_{11a} , LP_{11b}).

(4) Could the fiber shaper also control the polarization of the output light?

Response: Yes, the fiber shape also has the capability of polarization control. We conducted a series of experiments to explore polarization control capabilities, following the approach outlined in the ref. [7]. In our experiments, we employed a few-mode fiber to emphasize the polarization control with a smaller number of modes. As shown in Response Fig. 6, we characterized the polarization-dependent output spatial profiles under various bending conditions of the fiber. Specifically, we used a 10- μm core step-index multimode fiber (Thorlabs FG010LDA) with a pumping condition at 700 nm wavelength, 10 nJ energy, and linear polarization at 0° . Distinct modal content with different polarization states is clearly demonstrated under different bending conditions. These results demonstrate the potential of the fiber shaper for controlling the polarization of MMF output light.

Initial condition

Different bending states

Response Figure 6: polarization-dependent spatial profiles under initial condition and different bending states. The initial output spatial profiles of 0° and 90° polarizations show random combinations of modes with a relatively high extinction ratio. The modal content and the extinction ratio of 0° and 90° polarizations can be changed under different bending states. The pumping condition was set at 700 nm wavelength, 10 nJ energy, and linear polarization at 0°. The intensity profiles of 0° and 90° polarizations are normalized to the common maximum value for each configuration.

(5) Can the fiber shaper mimic the functionality of a fiber piano, serving as both tunable bandpass filter and dual-band filter?

Response: Yes, the fiber shaper can potentially serve as tunable bandpass filters [8] but will need further customization. Specifically, the fiber piano work, from our understanding, might have been designed to optimize the signal-to-background ratio (SBR) for bandpass filter performance in the linear regime. This high SBR could have been enabled by a locally bending fiber piano device and the spatial filtering of the output via splicing a single-mode fiber (SMF) to the multimode fiber (MMF). This additional step of MMF-to-SMF collection provides spatial filtering that may enhance the SBR at the cost of the total energy loss. In contrast, our approach prioritizes the enhancement of the energy within the desired bands with low loss and high peak power for nonlinear imaging. We designed the parameters of the fiber shaper to avoid significant loss, thus it is hard to serve as a tunable bandpass filter with high SBR with this set of design parameters.

However, we think this might be possible with a future fiber shaper with a smaller bending radius. By reducing the bending radius, higher loss [1] and stronger mode coupling (Response Fig. 1) can be induced. Consequently, we expect higher spectral tunability accompanied by greater energy loss, leading to more distinct spectrum distributions with higher SBR. Increasing the fiber length can result in higher mode-dependent loss [1] and narrower transmission spectral peaks [8]. Therefore, there is potential for achieving a tunable bandpass filter with suitable adjustment of the parameters.

(6) The authors exclusively examine the 30cm length of MMF. What prompted the selection of this specific length, and does it represent an optimal length that maximizes tunability?

Response: The selection of 30-cm long fiber is the outcome of the tradeoff between the pulse duration preferences of multiphoton microscopy (MPM) and the design constraints of the fiber shaper device. On one hand, a shorter fiber is beneficial for maintaining a short temporal pulse duration in MPM for efficient multiphoton excitation [2]. As demonstrated in Response Fig. 2, when we increased the fiber length from 30 to 40 cm, a significant pulse broadening occurs, which is unfavorable for high-order nonlinear processes (e.g., three-photon absorption). On the other hand, a short fiber poses a challenge to the system design due to the limited space for the fiber shaper, as each actuator takes a space of around 2 cm to implement in the current setup. More details on the choice of the number of actuators can be found in Response to Reviewer 1 Comment 2.

In summary, considering the balance between shorter pulse duration and tunability, we chose the fiber length of 30 cm. This configuration allows for effective fiber output customization without significantly compromising the short pulse duration that is needed for higher-order multiphoton imaging. However, this can be further optimized. For example, if we can reduce the size of each actuator and fit more actuators within smaller fiber lengths, we could produce more tunability without compromising the output pulse duration.

Accordingly, we have added a brief explanation about the choice of the specific fiber length in Methods Section 4.1.

Response Figure 2: Representative results of the comparison of pulse durations out of straight step-index multimode fibers with two different fiber lengths. The measurements were performed on the bandpass filtered (1100 ± 25 nm) output with 1040 nm pump as an example.

(7) The authors conducted 3125 different measurements by adjusting the fiber shaper, as demonstrated in Fig. 1(b). The algorithm or the automated search methodology employed for these extensive tests should be elucidated. Maybe if the code was open source, it could have more impact. Additionally, what was the average calculation duration to obtain a single optimal output?

Response: We thank the reviewer for pointing out the missing information about the algorithm's details. We used exhaustive search and greedy search for spectral optimization, and SNR optimization, respectively. Because the relatively low-dimensional control degrees of freedom (multi-point macro-bending) can introduce high degrees of freedom in the output space (spatial, temporal, and spectral), we noticed that we could have effective output modulation with a relatively small control input space and this can be readily achieved with using very simple optimization/search methods. With an increasing number of actuators/control variables, we anticipate a more intricate optimization algorithm is needed for a wider variety of objectives for simultaneous spatial, temporal, and spectral optimization.

For spectral customization (Fig. 1b), we used exhaustive search because of practical convenience (the spectrometer measurements can be easily synchronized with fiber shaper in real time, and input control space is small enough to finish exhaustive search of 5^5 states within 50 minutes. ~ 1 second acquisition time for spectrometer). This approach involves systematically going through all possible positions (states) of each actuator. It explores all combinations of N^n states, where N represents the number of actuators, and n represents the number of states that each actuator can operate in (although n can vary for different actuators). For spectral customization shown in original Fig. 5b (now Fig. 4b), the average calculation duration to obtain a single optimal output is about 50 minutes.

In the experiments of optimizing the multiphoton imaging signal-to-noise ratio (SNR), we implemented a semi-automated greedy search algorithm. This approach involves fixing all other actuators' positions and incrementally adjusting one actuator's state with high resolution (small step size) while continuously monitoring the imaging SNR. When a local maximum of imaging SNR is reached, we fix the state of that actuator and proceed to the next actuator in a similar manner. It took approximately 10 minutes to obtain a single optimal imaging SNR under these semi-automated optimization conditions. Each iteration was subjected to a delay of several seconds, primarily due to the logistical waiting time for the coordination between imaging acquisition on our imaging PC and the fiber control on another PC. For the experiments done in this work, we performed one-time optimization for one imaging session, so such waiting time was not an issue. However, if real-time adaptive optimization is needed in the future, synchronization between hardware will speed up this process significantly (the bottleneck will become the imaging time). Also, we would like to note that such optimization process likely yields a local optimal output, and we cannot guarantee global optimization using this method. We are actively working on further refining the algorithm to enhance its optimization capabilities.

While we recognize the potential benefits of open-sourcing our codes, we would like to clarify that the underlying optimization algorithm is not novel. We will share it here as supplementary materials but maybe not sophisticated enough for a github page.

We have added a brief explanation about the 3125 states and automated search methodology in the caption of Fig. 1. We have also attached the lines of codes in the file 'actuator_control.py' and 'actuator_running.ino' as supplementary materials.

(8) Are there specific predetermined actuator positions that result in optimal outputs for specific wavelengths, like maximizing the spectrum at 1800 nm or 2000 nm?

Response: We haven't observed such patterns in our experiments. With single mode fibers, it has been shown that a predetermined pulse property can produce solitons at certain predicted wavelengths. We also show in original Fig. 7 (now Fig. 6), that a predetermined LP₀₇ wavefront at the input can produce significantly more broadening than the weakly focusing one due to mode-dependent ZDW matching. However, we have not observed analogous trends (predetermined actuator positions for output properties) for fiber shaper applied to the multimode fibers (MMFs) when many modes are involved. This could be due to 1) the inherent complexity of the spatiotemporal nonlinear effects in MMFs, and 2) bending introduces intermodal coupling which further complicates the overall pulse propagation.

(9) Sections 2.2 and the initial part of Section 2.3 appear to consist of standard tests without the utilization of the fiber shaper, relying on older theories and results. The conclusions drawn from these sections appear relatively weak and do not contribute significantly to the core message of the paper. It might be beneficial to condense these sections and transition more swiftly to the primary topic, which is the tunability of the fiber shaper.

Response: We agree with the reviewer's suggestion. We have now moved the technical description of supercontinuum generation to Supplementary Materials and condensed Section 2.1-2.3 for a better focus on our core message.

(10) Figure 4(f) displays only 5 modes, yet the authors made use of the first 6 modes in their analysis.

Response: We thank the reviewer for noticing this. We used the first 6 modes in the analysis, but only 5 modes were shown in Figure 4(f). We have corrected in the original Fig. 4f (now Fig. 3f) to display the first 6 modes.

(11) In line 238, regarding the damage threshold of the GRIN fiber, I recommend the authors consider referencing (ref.2), here this threshold has been recently studied.

Response: We have added ref.2 in Results Section 2.2 as reference 67 to provide a more robust foundation for our discussion on the damage threshold of the GRIN fiber. The multiphoton ionization (MPI) effects provide a valuable insight into investigation of the laser-induced damages in the MMFs.

(12) In reference to the mode-content-dependent soliton spectral shifting discussed in lines 285-287, I suggest that the authors include a reference to ref.3. Its Authors have demonstrated that, in the presence of multimode solitons with identical energy but different initial mode components, there are notable differences in the resulting soliton spectrum. This observation could provide valuable insights into the substantial increase in the fold number (166-fold) observed within the soliton regime.

Response: We have incorporated ref.3 in Results Section 2.3 as reference 70 in the discussion of mode-content-dependent soliton spectral shifting, highlighting the reported differences in the soliton spectrum due to varying initial mode components. The demonstrated multimode soliton behaviors are consistent with what we observed in our experiments. The bending-induced mode coupling would lead to a different combination of modal content which will reflect on the soliton spectra as the frequency shift. Therefore, the soliton regime showed a high fold number of band energy enhancement and suppression.

(13) Line 418: In line 418, could you provide insights into how the authors determined that the input mode corresponds to LP₀₇? Additionally, in the context of the high-power regime, could you elaborate on the mode distribution at various main spectral peaks?

Response: The selection of LP₀₇ mode was to match the zero-dispersion wavelength (ZDW) of the fiber (Supplementary Fig. 8), which is crucial for spectral broadening with a different pump wavelength than 1300 nm (in this case, 1040 nm). We used a spatial-light modulator (SLM) to excite the LP₀₇ at the input, based on the method outlined in [9]. To determine that the input mode

corresponds to LP_{07} , at low-power levels (linear regime), we captured the near-field intensity profile of the fiber output in real-time, and maximized its correlation with theoretical intensity profiles of the LP_{07} by fine-tuning the phase pattern in the SLM. The correlation coefficient could typically reach 0.9 (1 is the theoretical maximum one can achieve).

The mode distribution under high-power conditions and its evolution across various main spectral peaks are shown in Response Fig. 7. For straight fibers, the spectrally-integrated (w.o. filter) output is predominantly the LP_{07} mode. At other wavelengths, the spatial profiles were mostly still LP_{0m} modes ($m \geq 7$). Shorter wavelengths tend to couple into higher-order LP_{0m} modes, while at longer wavelengths we observe lower-order LP_{0m} modes. The trend in this less-speckled mode distribution correlates with the distinct differences in effective indices (n_{eff}) between the LP_{0m} modes and their nearest LP_{1m} counterparts, as discussed in [10]. In contrast, the curved fiber presents a different mode distribution, where speckled patterns become more pronounced. This change is attributable to the strong disorder and mode coupling induced by the fiber shaper.

Response Figure 7: Representative results of the measured near-field intensity spatial distribution at various spectral bands under the excitation condition of LP_{07} pump. Two sets of data are provided: the upper row shows results for a straight fiber, while the lower row shows those for a curved fiber. Each intensity profile is normalized to its own maximum value for visualization purposes. The associated spectra are shown in the lower panel where the straight fiber is adapted from the manuscript.

(14) On page 16, line 319, the Authors should mitigate their claim “To the best of our knowledge this is the broadest continuous supercontinuum (560-2200 nm)...” by referencing to earlier literature where similar or even broader SCG was demonstrated using either GRIN or SI MMFs (refs.4-6)

Response: We appreciate the reviewer's suggestion regarding the rigor of the claim about the supercontinuum (SC) we achieved. Our claim was intended to highlight the performance under the specific ultrafast femtosecond pumping conditions we employed. To clarify this and acknowledge the accomplishments in the field, we have removed the use of “broadest” and adjusted our statement to reflect the context more accurately and cite the relevant works. We have added refs.4-6 in Results Section 2.3 as references 10, 71, and 21.

(15) On page 2, line 46, when mentioning nonlinear frequency generation with MMF, ref.7 which demonstrated frequency conversion via geometric parametric instability should be cited.

Response: We have cited ref.7 in Introduction Section 1 as reference 4, to acknowledge the demonstrated frequency conversion via geometric parametric instability in MMF.

(16) It is suggested to add a couple of additional important review papers on nonlinear effects in MMF (refs.8-9)

Response: We have added more additional important review papers including refs.8-9 in Introduction Section 1 as references 2 and 3, where they are discussed in the manuscript to highlight the advancements in the field of nonlinear effects in MMF.

Typos:

- line 13: “Multimode fibers (MMFs) have recently reemerged as an attractive avenue”
⇒ “Multimode fibers (MMFs) have recently reemerged as attractive avenues”
- line 34: “have become a renewed focus of interest” ⇒ “have become renewed focuses of interest”
- line 60: “ignite advances of cutting-edge applications,” ⇒ “ignite advances in cutting-edge applications,”
- line 90: “in linear and quantum regime” ⇒ “in linear and quantum regimes”
- line 99: “were launched in to” ⇒ “were launched into”
- Fig. 3: “the spectrum and beam profile at each input pulse energy is normalized to their”
⇒ “the spectrum and beam profile at each input pulse energy are normalized to their”
- Line 163: “The discrepancy in distribution of power spectral density” ⇒ “The discrepancy in the distribution of power spectral density”
- line 338: “mercial laser source based on optical parametric amplifier” ⇒ “mercial laser source based on an optical parametric amplifier”
- line 354: “15-fold, 11-fold and, 6-fold” ⇒ “15-fold, 11-fold, and 6-fold”
- Fig. 7: “were launched into the the same standard silica” ⇒ “were launched into the same standard silica”
Fig. 7: “Inset shows the spectal tunability between” ⇒ “Inset shows the spectral tunability between”

List of references

1. N. Berti, et al. Interplay of Thermalization and Strong Disorder: Wave Turbulence Theory, Numerical Simulations, and Experiments in Multimode Optical Fibers. Phys. Rev. Lett.129, 063901 (2022). <https://doi.org/10.1103/PhysRevLett.129.063901>
2. M. Ferraro, et al., "Multiphoton ionization of standard optical fibers," Photon. Res. 10, 1394- 1400 (2022). <https://doi.org/10.1364/PRJ.451417>
3. Yifan Sun, et al., "Multimode soliton collisions in graded-index optical fibers," Opt. Express 30, 21710-21724 (2022). <https://doi.org/10.1364/OE.459447>
4. K. Krupa, et al., "Spatiotemporal characterization of supercontinuum extending from the visible to the mid-infrared in a multimode graded-index optical fiber," Opt. Lett. 41, 5785-5788 (2016) <https://doi.org/10.1364/ol.41.005785>
5. S. Perret, et al, "Supercontinuum generation by intermodal four-wave mixing in a step-index few- mode fibre," APL Photonics 1 February 2019; 4 (2): 022905. <https://doi.org/10.1063/1.5045645>
6. Z. Eslami, et al. Two octave supercontinuum generation in a non-silica graded-index multimode fiber. Nat Commun 13, 2126 (2022). <https://doi.org/10.1038/s41467-022-29776-6>
7. K. Krupa et al., "Observation of geometric parametric instability induced by the periodic spatial self-imaging of multimode waves," Phys. Rev. Lett. 116, 183901(2016). <https://doi.org/10.1103/physrevlett.116.183901>
8. K. Krupa, et al., "Multimode nonlinear fiber optics, a spatiotemporal avenue," APL Photonics 1 November 2019; 4 (11): 110901. <https://doi.org/10.1063/1.5119434>
9. L. G. Wright, et al. Physics of highly multimode nonlinear optical systems. Nat. Phys. 18, 1018– 1030 (2022). <https://doi.org/10.1038/s41567-022-01691-z>

Response: We sincerely thank the reviewer for their thorough reading of our manuscript, pointing out these typos, and providing this list of references. We have addressed each of the mentioned typos and make corresponding corrections in the revised manuscript.

Reviewer #3 (Remarks to the Author):

1. The paper deals with the control of multimodal nonlinear effects at high peak power levels: by leveraging not only the spatial but also the temporal degrees of freedom in step index MMFs using a programmable fiber shaper. This method represents from the point of view of the authors the first method that enables simultaneous access to the spatial and temporal degrees of freedom of multimodal nonlinear pulse propagation, achieving high tunability and broadband high peak power. Its potential as a nonlinear imaging source is further demonstrated by applying the MMF source to multiphoton microscopy, where widely tunable two photon and three-photon imaging is achieved with adaptive optimization.

Response: We thank the reviewer for the summary. We have provided the point-by-point response below to address the individual comments with a number index added to enhance the scientific exchange.

2. Spatio-temporal analyses of propagation in multimode fibers have already been published and should be cited. As an example, the paper by K. Krupa et al. "Observation of geometric parametric instability induced by the periodic spatial self-imaging of multimode waves," Phys. Rev. Lett. 116, 183901 (2016); should be included as an example of a new nonlinear conversion process observed in optical multimode fiber which can drastically change the supercontinuum generation process.

Response: We appreciate the reviewer for pointing out the relevant study in this field. We have added this paper in Introduction Section 1 as reference 4.

3-1. In addition, another paper on temporal shortening in multimode fibers also published by Krupa et al. "Spatiotemporal light-beam compression from nonlinear mode coupling," Phys. Rev. A 97, 043836 (2018)" should also be cited as an early example of temporal versus spatial modal evolution.

Response: We have added this paper in Introduction Section 1 as reference 5 as examples of existing studies on nonlinear effects in multimode fibers.

3-2. The paper by Y Leventoux et al. "3D time-domain beam mapping for studying nonlinear dynamics in multimode optical fibers", Optics Letters 46 (1), 66-69 (2021), demonstrates the complex spatiotemporal interweaving of temporal and spatial effects in multimode fibers with nonlinear conversions.

Response: We have added it in Introduction Section 1 as reference 6 as examples of existing studies on nonlinear effects in multimode fibers. The proposed 3D spatiotemporal mapping in this recommended paper is a powerful analysis tool and we will consider using it in our future studies.

4. Other papers on supercontinuum in multimode fibres have also been widely published and some references should be cited:

- Shalaby, B. M et al. "Visible Supercontinuum Generation by Dual-Wavelength Pumping in Multimode Rectangular Optical Fibers," CLEO: Science and Innovations, Paper# STh3N.4 (2014)
- Zhou, Renlai et al. "Half mJ Supercontinuum Generation in a Telecommunication Multimode Fiber by a Q-switched Tm, Ho:YVO4 Laser," Journal of the Optical Society of Korea 19(1) 7-12 (2015)
- Lopez-Galmiche, G. et al. " Visible supercontinuum generation in a graded index multimode fiber pumped at 1064 nm," Optics Letters 41(11) 2553-2556 View: (2016)
- Krupa, K. et al. "Spatiotemporal characterization of supercontinuum extending from the visible to the mid-infrared in a multimode graded-index optical fiber," Optics Letters 41(24) 5785-5788 (2016)
- Eftekhar, M. A. et al. " Versatile supercontinuum generation in parabolic multimode optical fibers," Optics Express 25(8) 9078-9087 See: (2017)

Response: We appreciate the reviewer's suggestions and added them in Introduction Section 1 as reference 7-9, and 10, as examples of existing studies on nonlinear effects in multimode fibers.

5. In this paper, the authors report successfully controlling nonlinear effects by exploiting spatial and temporal degrees of freedom through programmable fiber shaping. This is clearly not the case from my point of view. They modified the curvature of the fiber, which affects the spatial and then temporal propagation of the short pulse. There is a big difference between modifying the spatio-temporal evolution and controlling the spatio-temporal profile. This must be said and explained correctly. As an example, there is no good control of the modal excitation, only a curvature of the fiber which creates a linear coupling between modes. This coupling is not controlled in order to select the number of modes or the modes desired for propagation.

Response: We understand the reviewer's concern and would like to clarify what we meant by spatiotemporal control in the original manuscript. In the body of work on pulse propagation in nonlinear media such as multimode fibers, spatiotemporal control usually targets three parts: (1) spatiotemporal control of the input (e.g., [11] for spatial and [12] for temporal) (2) spatiotemporal control of the in-medium propagation dynamics [13], [14], and (3) spatiotemporal control of the output field [15], [16]. As the reviewer mentioned, these three targets mean different things, despite all sharing the word "spatiotemporal control".

In this work, our spatiotemporal control targets the control of the in-fiber nonlinear pulse propagation dynamics by introducing programmable time-dependent disorder along the fiber [4]. This method directly accesses the temporal degree of control of nonlinear pulse propagation (nonlinear effects), as well as the spatial aspect. The resulting change of the output field (original Figs. 5 and 6, now Figs. 4 and 5) is the desired consequence of controlling the nonlinear effects,

as pointed out in [11], [17]. Specifically, the temporal degree of freedom of the pulse propagation is provided by introducing a time-dependent disorder [4], a mechanism implemented through location-dependent bending using the fiber shaper. The rationale lies in the fact that a single short pulse arrives at various local curvatures at distinct time instances, as illustrated conceptually below (adapted from Fig. 1a in the original and revised manuscript). More details of the process of translating location-dependence into time-dependence can be found in [4].

Response Figure 8: Conceptual illustration of the direct access to the temporal degree of freedom to tailor the nonlinear pulse propagation using our method. The time-dependent disorder is implemented as position-dependent macro-bending, related by $t_i = l_i/v_g$ (l_i the axial position of the i -th curvature) for a single fiber mode. Adapted from Fig. 1a in the original and revised manuscript.

Regarding the specific concern on accurate spatial (or temporal) control of the input, previous work demonstrated that it is possible to change the fiber output by spatial control of the input beam [3], [11], [17]–[21]. Our proposed approach of controlling the multimodal nonlinear propagation, instead of the input, is an alternative method to achieve customization of the output field with potentially more degrees of freedom, due to its analogy to the concept of Multiple-Plane Light Conversion (MPLC) (see our Response to **Reviewer 1 Comment 3** for more details in this regard). We did not aim to control the input in this work by wavefront shaping or pulse shaping, although it is compatible with controlling the evolution and we are exploring it in the next study to further expand the control degrees of freedom. In our experiments, instead of controlling modal compositions at the input [3], [11], [17]–[21], the modal compositions change in our experiments as a result of controlling the time-dependent disorders. To visualize this change, we have added an illustrative experiment using a few-mode fiber as shown in Response Fig. 5, and copied below. It also provides the most straightforward evidence that the fiber-shaper-induced “coupling”, which results in energy exchange between modes and subsequent variations of the modal compositions, can be customized.

Response Figure 5: Normalized output spatial profiles of a 10- μm core step-index multimode fiber under various bending conditions. Each profile is individually normalized. The initial output beam profile shows a mixture of approximately three linearly polarized (LP) modes (LP_{01} , LP_{11a} , LP_{11b}).

However, we understand the potential concerns of the reviewer. To avoid unnecessary mix-up on the usage of “spatiotemporal control” by future readers, we have made our definition of “spatiotemporal control” clearer in the revised manuscript (Abstract, Introduction Section 1, Fig. 1a caption, and Discussion Section 3). Also, to avoid distraction from the core message, we have revised our title to: Spectral-temporal-spatial customization via modulating multimodal nonlinear pulse propagation.

6-1. Are the authors able with this system to excite one or two given modes with temporal control of the duration of the input pulse?

Response: As described in the response to the previous comment, the spatiotemporal control here specifically targets the pulse propagation [13], [14], not temporal control of the input pulse. To avoid unnecessary mix-up on the usage of “spatiotemporal control” by future readers, we have made our definition of “spatiotemporal control” clearer in the revised manuscript (Abstract, Introduction Section 1, Fig. 1a caption, and Discussion Section 3). For a more detailed discussion please refer to the Response to Comment 5 above.

6-2. Are the authors able with this system to excite only high order modes with control of the pulse profile?

Response: We merged the response to this question with the next one as they are highly related.

6-3. Are the authors able with this system to excite in output only low order modes by fixing the spectrum profile and a precise pulse duration?

Response: As described above in the Response to Comment 5, the spatiotemporal control here specifically targets the pulse propagation [13], [14], not spatial control of the modal compositions at the input [11]. We have added clarifications on this point in the revised manuscript. For a more detailed discussion, we kindly direct the reviewer to our Response to Comment 5.

However, if the reviewer is curious about the change of the output pulse properties as a result of the time-dependent disorder, we have provided an illustrative example of the modulation on the output modal composition shown in original Fig. 5 (now Fig. 4), original Fig. 6 (now Fig. 5), and Response Fig.5.

7. In the time domain, there is no control of the coupled pulse in the fiber. No control over the shape or duration of the pulse is clearly explained. Only propagation modification using actuators is used. It is then evident that due to the additional bending and stress on the fiber, the energy distribution across the transverse modes will be changed, which directly changes the time domain due to the short pulse used in the experience. Beyond a modification of the experimental propagation conditions, no control is really demonstrated. Can the authors show the modal content based on the actions of the actuators? For example, can authors impose pulse duration and spatial mode content on their actuators?

Response: We feel this comment may share similar concerns as Comment 5, in terms of the definition of the term “spatiotemporal control”. Please refer to Response to Comment 5 for a more detailed discussion.

Briefly, the spatiotemporal control in this work refers to the time (position)-dependent control of the multimodal pulse propagation. Each actuator introduces modulation to the modal interactions (spatial) at different temporal instances (temporal) during pulse propagation, rather than the input pulse itself. If the reviewer is curious about the spatial and temporal properties of the output pulses, the mode content of the fiber output based on the actions of the actuators is shown in Response Fig. 5, demonstrating effective modulation on the modal content using the fiber shaper. Simultaneous customization of modal content and pulse duration of the output pulse is also attainable, which has been demonstrated in the original Fig. 5c (now Fig. 4c).

To avoid unnecessary mix-up on the usage of “spatiotemporal control” by future readers, we have made our definition of “spatiotemporal control” clearer in the revised manuscript (Abstract, Introduction Section 1, Fig. 1a caption, and Discussion Section 3). Also, to avoid distraction from the core message, we have revised our title to: Spectral-temporal-spatial customization via modulating multimodal nonlinear pulse propagation.

8. Authors claim line 68 that they introduce a new way of controlling multimodal nonlinear effects. How it is possible without controlling the modal content of the beam and the initial pulse duration and shape?

Response: As described above in the Response to Comment 5, the spatiotemporal control here specifically targets the pulse propagation [13], [14], not spatial control of the modal compositions or the temporal control of the pulses at the input [11]. We have added clarifications on this point in the revised manuscript. For a more detailed discussion, we kindly direct the reviewer to our Response to Comment 5.

Briefly, in this work, our goal is to customize the fiber output field, which essentially stems from the effective control of the nonlinear effects. The nonlinear effects in MMFs occur between modes [17] and can be potentially controlled by tailoring the intermodal interactions. This can be achieved through either control of the input modal composition (original Fig. 7, now Fig. 6), the control of the pulse propagation (the proposed core method of this work, Fig. 1), or a combination of both (original Fig. 7, now Fig. 6). Previous work demonstrated that it is possible to change the fiber output by spatial control of the input beam [3], [11], [17]–[21]. Complimentary to input control approaches, the novelty of our approach lies in introducing programmable time-dependent disorders to modulate multimodal nonlinear pulse propagation, so that the spectral-temporal-spatial properties of the output field can be customized with high degrees of freedom to suit diverse applications.

9-1. They claim also (line 66) that they realized, in high-power regimes by exploiting not only the spatial but also the temporal degrees of freedom a control of the output beam.

Response: As described above in the Response to Comment 5, the spatiotemporal control here specifically targets the pulse propagation [13], [14], not direct spatial or temporal control of the output or the input [11]. Please refer to Response to Comment 5 for a more detailed discussion.

Briefly, as mentioned in the original text, the spatiotemporal control is applied to the multimodal nonlinear pulse propagation (nonlinear effects), not the input or directly the output, which results in effective customization of the spectral-temporal-spatial properties of the output fields to suit wavelength- and power-demanding applications.

9-2. They claim also to control of the brilliance of the output beam by never showing a measure proving that evolution. What is the M^2 coefficient of the output beam versus activating actuators?

Response: In the original manuscript (lines 30, 57, 58, 89, 211, 332, and 407), we used “spectral brilliance” to represent the energy distribution within the spectral band. Our results show that the energy of desired spectral bands can be enhanced or suppressed (original Figs. 5a and 5b, now Figs. 4a and 4b) and customized according to the desired criteria (original Fig. 6, now Fig. 5).

However, we understand there could be potential misunderstanding surrounding the term “spectral brilliance”, as pointed out in [22].

To avoid potential misunderstanding from future readers, we have replaced "spectral brilliance" with “spectral brightness” to improve the clarity in the revised manuscript.

Regarding the M^2 analysis, the MMF output fields, as shown in Response Fig. 9 (adapted from original Fig. 5c, now Fig. 4c), are usually highly speckled under varying actuator configurations (except for in the near-infrared-II region). Therefore, the M^2 analysis may not be an applicable metric to evaluate the beam quality and thus is not provided.

Response Figure 9: Near-field intensity fields at the fiber output, under varied actuator configurations. Adapted from original Fig. 5c (now Fig. 4c).

10. To me, this paper is just a demonstration that short pulse coupling in MMF can be modified along the propagation by manipulating the curvature at many propagation distances. Is it possible with this system to shorten or increase the pulse duration in relation to the modal content? Is it possible to precisely control modal content?

Response: More details on what this work is about and the scope of control can be found in our Response to Comment 5.

Briefly, this is the first method that can access both the spatial and temporal dimensions of the pulse propagation in multimode fibers (MMFs), achieving effective customization over the spectral-temporal-spatial properties of the output pulses (existing work [3], [11], [17]–[20] mostly demonstrated the spectral property). We further demonstrated that the imaging quality of multiphoton microscopy with multimode fiber can be boosted by adaptively optimizing the fiber shaper. It is correct that this idea is implemented by engineering the curvature at different propagation distances and demonstrated as a 30-dollar 3D printed fiber shaper. However, the simplicity of this method, if not an advantage, does not mean it is not novel or effective. Regarding the question on controlling the modal content, we did not intend to control at the input; at the output, we kindly refer the reviewer to Response Figs. 5 and 9.

Indeed, arbitrary control over individual properties of the output fields can be done with wavefront shaping and pulse shaping [23]. However, arbitrary control of MMF output remains challenging due to the inherent complexity of the spatiotemporal nonlinearities in MMFs, despite the rapid advances of multimode fiber nonlinear optics theory and experiments. Some existing works [3], [11], [17]–[20] have demonstrated controlling the spectral property, whereas in this work we have demonstrated customization of the output pulses in the spectral, temporal, and spatial domains, which benefits a breadth of applications, including more efficient multiphoton signal generation as demonstrated in our work. Meanwhile, it is worth noting that our goal here is to develop a tunable and adaptive MMF source for wavelength- and power-demanding applications, e.g., multiphoton microscopy (MPM). While the precise control of individual properties is beyond the scope of this work as our focus is more application-driven, it will be interesting to keep pushing advanced control of multimodal nonlinear optics in the future.

11-1. The authors state at line 111 that: “By producing local index perturbations, the actuators cause an energy coupling between modes (39, 40) at several points in the evolution of the pulse. This change does not prove a control of the energy exchanges between modes and a control of the characteristics of the pulses.

Response: When the “perturbation” (i.e., time-dependent disorder) is applied, the spatial, spectral, and temporal properties of the output pulses change accordingly as a result of local refractive index changes (original Fig. 5, now Fig. 4). As for the control of the energy exchange, the distinct spatial intensity patterns (initial and optimized) in original Fig. 6 (now Fig. 5) demonstrates energy exchange between modes. If the reviewer is concerned about arbitrary control of the output or the input, we refer to the response to the previous comment.

11-2. What means “at multiple time points”? Is it at different positions in fiber? If so this is not a temporal control along the pulse duration.

Response: In line 111 of the original manuscript (now line 120), “at multiple time points” means at multiple temporal instances along the pulse propagation, which in our practical implementation is equivalent to “at different positions in fiber”. This position-dependent macro-bending is the implementation of time-dependent disorder, and it can directly access the temporal dimension along the pulse propagation. Our “temporal” targets tailoring the in-fiber multimodal pulse propagation by introducing time-dependent disorder, which is different from the temporal control of the input (e.g., pre-chirping or pulse shaping [12]) or output pulse [15], [16]. For more details see our Response to Comment 5.

12. In Figure 1 the authors show different spectra depending on the actuations of the actuators. Is it possible to start from the spectral profile we need and manipulate the actuators to obtain the result?, with a given pulse shape and duration?

Response: Figures 6j-6r in the original manuscript (now Figs. 5j-5r) may have described what the reviewer asks. However arbitrary control of spectral intensity, pulse shape, and duration is not achieved at this point with this method. For more details please see our Response to Comment 10.

13-1. The authors, in line 130, stated that solitary waves consisting of multiple spatial modes begin to form...” Can the authors give a proof of the soliton contents? Is it possible to control this soliton content with the actuators?

Response: Yes, the multimode solitons (MMSs) have been observed in multimode fibers (MMFs) by multiple groups. The MMSs consist of multiple distinct but temporally-aligned spatial modes [24]–[27]. Given our excitation condition by weakly focusing, the MMSs are anticipated to be formed by a few lower-order modes [24], [26], [27]. Response Figure 10 (adapted from the original Figs. 3 and 5c (now Supplementary Fig. 1 and main Fig. 4c)) below presents representative characterization results of the spectrally-filtered (1900 ± 100 nm) output of a straight fiber pumped with an 800-nJ input energy; the corresponding spectrum is also shown for reference. It is evident that (1) the field consists of only a few low-order modes, (2) there are no discernible secondary temporal structures in the autocorrelation signal, and (3) the spectrum consistently undergoes red-shifting with higher input energy (i.e., Fig. 3a in the original manuscript, now Supplementary Fig. 1a). All confirms that the field obtained after the filter 1900 ± 100 is MMS.

Response Figure 10: Representative experimental results of the spectrally filtered fiber output fields at an input energy of 800 nJ. Adapted from Figs. 3 and 5c in the original manuscript (now Supplementary Fig.1 and main Fig. 4c).

To gain more insights into the MMS in step-index MMFs, we performed numerical simulations based on the method outlined in [28] (see Methods for more details). The results are shown below. The pulse energy (250 nJ) is evenly distributed into the first six spatial modes (Mode 1 to Mode 6); two additional high-order LP_{0n} modes (Mode 7 and 8) are also included in the simulation. Other parameters remain the same as in the original manuscript. Supplementary Figure 11a visualizes the MMS formation during propagation. In Supplementary Fig. 11b, it is clear the eight

modes have the same temporal delay, profile shape (Gaussian-like), and duration. The 70-fs temporal FWHM of the MMS in simulation is in good agreement with the 111-fs autocorrelation FWHM in experiments. In short, our experimental and simulation results collectively confirm the formation of MMS.

Response Figure 11: **a**, Beam propagation in the temporal and spectral domains, showing the formation process of the MMS. **b**, Spectrally-filtered (1900 ± 100 nm) mode-resolved temporal profiles, inset shows the time-integrated beam spatial pattern.

Regarding controlling the soliton, yes, our method can. We kindly refer the reviewer to Response Fig. 4 and “ 1900 ± 100 nm” in original Fig. 5c (now Fig. 4c), where it shows that the mode composition can be tuned under different actuator configurations, along with a slight change in the autocorrelation signal width. We note in some cases secondary temporal structures show up, this is likely due to the fixed spectral width of the bandpass filter (1900 ± 100 nm) which may under- or over-collect the most redshifted MMS.

13-2. What are the losses introduced by the actuators on the propagating output beam and on each modes.

Response: The total transmission loss induced by the actuators has been measured as 2% at 800-nJ input pulse energy (Results Section 2.1 for more details). If the reviewer is curious about the overall mode content change, we have presented simulation results of the actuator (bending)-induced mode coupling for the first 6 spatial modes in the original Fig. 4 (now Fig. 3) and simulation results for the first 55 spatial modes in Response Fig. 1 (now added as Supplementary Fig. 10); experimental results using a few-mode fiber is summarized in Response Fig. 5.

13-3. Can the authors give a coupling coefficient between transverses modes by using actuators?

Response: The coupling coefficients have been provided in the original manuscript (the original Figs. 4e and 4j, now Figs. 3e and 3j). Additionally, we have now added simulation results of the coupling coefficients for the many-mode (the first 55 spatial modes) case in Response Fig. 1 and experimental results of a few-mode fiber in Response Fig. 5.

14. The figure 3 show a standard supercontinuum obtained at the fiber output. What is the novelty here?

Response: We understand that the supercontinuum results are not surprising to anyone in the fiber nonlinear optics community. The power evolution map was provided to illustrate the different mechanisms responsible for the broadening, which lays the foundation for the mechanism-dependent tunability of the output pulse properties in the subsequent sections. In addition, nonlinear effects in step-index multimode fibers (MMFs) are not as popular compared to graded-index MMFs. These results provide reference value to the scientific community in conjunction with other work on step-index MMFs [24], [26], [27], [29]–[32]. However, we understand the reviewer’s concern that this might distract readers from our core message. To better focus on the core message of this work, we decided to move Fig. 3 and the corresponding discussion in Section 2.2 to Supplementary Note 1 and Fig. 1.

15. In Section 2.3, the authors claim that actuators modify the local refractive index profile, causing energy coupling between the modes at several points in time during the evolution of the pulse, which allows in total to high-dimensional spatio-temporal control of multimode nonlinear pulse multiplication”; Where is the control here. For example, is it possible to fix the coupling between modes LP01 and LP21 or other pairs of modes?

Response: This is highly related to Comments 5 and 10, in terms of the definition of spatiotemporal control, and arbitrary control of the output field. We kindly direct the reviewer to our responses to Comments 5 and 10 for more details.

16. The authors claim again line 200 that the spatial and temporal degrees of freedom of nonlinear pulse propagation can be simultaneously controlled. This is not true. They are changing the linear coupling between modes which affect the pulse shape, but no control is really shown here. As an example is it possible to shorten the pulse duration or to choose a given pulse duration with energy only on the LP01 mode?

Response: This is highly related to Comments 5 and 10, in terms of the definition of spatiotemporal control, and arbitrary control of the output field. We kindly direct the reviewer to Responses to Comments 5 and 10 for more details.

17. The authors claim also line 238 that: “ Apart from the tunability, we also observed that the GRIN MMFs exhibit a lower threshold for laser-induced damage. As we gradually increased the input pulse energy for the GRIN fiber, we observed a decrease in total output power and a reduction in the output spectral span, suggesting the occurrence of fiber damage. This lower damage threshold is possibly due to the parabolic index profile which leads to more severe self-focusing effects (56), and the Germanium dopant in the GRIN fiber core (57).”

This statement is obvious and everyone says that the modal area of lower order modes is different in step and grin fibers.

Response: Yes, the reviewer is correct that it is well-known that graded-index multimode fibers exhibit a smaller modal area than step-index multimode fibers, which is also shown in Figs. 4c and 4h in the original manuscript (now Figs. 3c and 3h). Our intention was to provide one of the experimental observations that motivated our exploration using step-index MMFs. It might be obvious to the reviewer but if it helps a junior researcher understand and think more on this aspect, we think it is ok to be slightly redundant.

In the case that the reviewer is curious about other factors of damage, the reduced modal area may not be the only factor contributing to laser-induced damage. The core material in the step-index fiber is undoped silica, which is different from the germanium-doped core of the graded-index fiber and may also lead to a distinctive damage threshold. This is a very interesting and open topic [32]–[34] and we are actively investigating the factors that affect the damage threshold and how to overcome it with collaborators.

18. The authors stated that: “The high peak power of broadband (approaching MW on average) was achieved by combining spectral energy reallocation (up to 166 times) and temporal shortening (up to 4 times). Such capabilities are only enabled by the fiber shaper, a single fully fiber modulator without the need for 3 external wavefronts.”

Just to show that this assertion is slight, it is enough to take the transformation of a normal pulse during the propagation in abnormal regime to show its transformation into solitons whose peak power can be very largely increased by the simple effect of the nonlinearity and dispersion. This statement is therefore true but very widely known without the authors putting this propagation regime into perspective of their experience.

Response: Certainly, we agree with the reviewer that in the anomalous dispersion regime, the interplay of nonlinearity and dispersion can lead to high-peak-power solitons, as illustrated in the Response to Comment 13-1 (soliton content) and our characterization results in the original Fig. 5 (now Fig. 4). However, in this work, we are more interested in achieving a broadband high-peak-power source for a wide range of applications. The soliton is spectrally limited to the anomalous dispersion regime (above 1300 nm for silica fibers), outside which the intricate interplay between the nonlinearity and dispersion significantly broadens the pulse duration, thus diminishing the output peak power. Our method, by modulating the modal interactions, can boost the peak power (e.g., shorten the pulse duration and/or increase the spectral band energy), resulting in a broadband high-peak-power source within and also outside the soliton regime. Another reason why we provide this propagation regime of soliton is because we observed highly wavelength-dependent spectral tunability (i.e., high in dispersive wave and soliton regime, and lower in nonlinear phase modulation regime) and temporal tunability (the other way around). We found this observation

intriguing and believe the investigation results are not only interesting to share but also informative for the community to design an even better fiber shaper or modulation method.

Reviewer #4 (Remarks to the Author):

The authors propose an intriguing method for achieving flexible spatiotemporal modulation of high peak power femtosecond laser light pulses using step-index (SI) multimode fibers (MMFs) through nonlinear optical processes. Despite the simplicity of implementing this method by bending the SI MMFs with a low-cost “fiber shaper”, it has achieved broad 2-octave spectral modulation at its maximum. The careful examination of the spatiotemporal modulation has demonstrated the advantages of this proposed method, including simplicity, broad spectral tunability, and high power tolerance, compared to previously reported methods using graded-index (GRIN) MMFs. While the section describing the profile of the modulated output has been thoroughly investigated, there is room for improvement in the section describing its application in multiphoton imaging. I hope that the comments below will be useful for the authors.

Response: We appreciate the reviewer’s positive comments and constructive suggestions. Here are our point-by-point responses.

Major comments

(1) The Methods section lacks a clear explanation of the multiphoton imaging setup. The spatial intensity profile of the output beam (shown in Fig6 l, o, r) does not exhibit typical distributions like Gaussian due to the influence of the MMFs and the modulations. Especially in that case, which spatial range of the beam is introduced to the objective pupil is important to evaluate the imaging quality because the spatial distribution of the beam influences on the PSF. I recommend the authors describe how large the beam diameter ($1/e^2$ radius) of the output laser light expanded by the relay optics (beam expander) inside the multiphoton microscopy and what proportion of the diameter of the output beam is introduced. Also, please provide information about the dichroic mirror and the parabolic mirror.

Response: We thank the reviewer for pointing out the missing information and the need for more clarification in our multiphoton imaging experiment. We have added the details of our setup to the Methods section, and the system schematic is shown in Response Fig. 12 (Supplementary Fig. 11). In brief, the fiber output is collimated by a parabolic mirror of 25.4-mm focal length (Edmund Optics 36-586), followed by a pair of galvanometer scanners and 4-f relay systems. The original collimated beam size is estimated as 5.5 mm, calculated from the focal length of the parabolic mirror and the half NA of the MMF ($NA = 0.22$), given that the output beam comprises a mixture of lower-order modes (smaller NAs) and higher-order modes (larger NAs). The collimated beam is slightly cropped by the first galvo mirror with an aperture size of 5 mm. The beam is then

expanded by a 4-f relay system, resulting in a diameter of 16.65 mm at the back focal plane of the objective lens which matches and is designed to slightly overfill the objective's pupil size (15 mm). A dichroic mirror (650-nm long pass filter) is placed before the objective lens, which separates the excitation from the emitted fluorescent signals in the detection channels.

Response Figure 12 (Added as Supplementary Figure 11): Schematic of the multiphoton imaging setup. PM: parabolic mirror; BPF: bandpass filter; DM: dichroic mirror; LP: long pass; OBJ: objective lens; PMT: photomultiplier tube.

We have incorporated the implementation details of our multiphoton imaging system in Methods Section 4.1 and Supplementary Fig. 11.

(2) In Fig. 6s, fluorescent images are quantitatively evaluated, and their intensity is compared with theoretical values. While the 2PF result showed a good match, the 3-photon applications did not. This discrepancy may arise from the lack of consideration for differences in the point spread functions (PSFs). Personally, I would prefer to estimate the PSFs using fluorescent beads below the diffraction-limited size, which could provide insights into the imaging quality of 3-photon applications compared to the 2PF, as the authors mentioned in L385-386.

Response: We appreciate the reviewer's advice on the PSF measurement and the quantitative evaluation of the discrepancy between the calculation and the experiment.

To provide insights into the imaging quality of the 3PF compared to the 2PF, we first used 0.1- μm beads (Bangs Laboratories FCDG002) to measure the PSFs for both 2PF and 3PF excitation with the multimode fiber (MMF) source, together with 3PF excitation with the laser (Cronus-3P) as the gold standard reference. As shown in Response Fig. 13, the 3PF excitation's PSF using the laser had a full width at half maximum (FWHM) of approximately 0.57 μm . For the fiber source, the

FWHM for the 2PF excitation was $1.52\ \mu\text{m}$ and for 3PF was $0.85\ \mu\text{m}$. The smaller PSF for the 3PF excitation can explain why less image blurring occurred compared to 2PF excitation as shown in Fig. 5.

Response Figure 13 (Added as Supplementary Figure 12): PSF measurement of the 3PF excitation with the laser (reference) and the 2PF and 3PF excitation with the fiber source. The size of the bead is $0.1\ \mu\text{m}$.

The predicted values in original Fig. 6s (now Fig. 5s) are calculated as the ratio of the predicted signal after optimization to that before optimization. The reason why the match between the theoretical and measured values of signal improvement is about $\sim 96\%$ for 2PF and $\sim 66\%$ for 3PF is because 1) the signal prediction equation we use, $S_n \propto E_p^n / \tau_p^{n-1}$ ($n = 2, 3$), does not take into account the PSF (i.e. assuming PSFs do not change); 2) unlike Gaussian beam imaging, the PSF with a multimode fiber output changes with different fiber shaper configurations, depending on the output modal content.

Original Fig. 6l, o, r (now Fig. 5l, o, r) show the dramatic changes in the spatial profiles (multimodal content) of the fiber output with respect to different fiber configurations. To further illustrate how fiber shaper configurations directly affect the imaging PSF, we have provided experimental data of the changing PSFs in relation to different fiber shaper configurations, as suggested by the reviewer. We started the experiment with $0.1\text{-}\mu\text{m}$ beads as shown in Response Fig. 13. However, we observed significant signal attenuation during the continuous scanning of the $0.1\text{-}\mu\text{m}$ beads, likely due to photobleaching, which makes it technically challenging to

continuously and reliably monitor the PSF variations during different bending conditions. This issue can be addressed by purchasing brighter beads.

For illustration purposes, we monitored how different bending configurations modify the PSFs using 1.0- μm beads (Bangs Laboratories FCDG006) to ensure a higher and more stable level of signals, as shown in Response Fig.14. While some configurations (e.g., an optimized fiber shaper configuration state 2 in 900-nm for 2PF and 1225-nm for 3PF) showed near-Gaussian PSFs, certain configurations (e.g., the initial fiber shaper configuration state 1) showed significantly different PSFs. In particular, 3PF can be more sensitive to the changes in the fiber output spatial profiles (Fig. 5l, o, r) than 2PF because of its higher nonlinearity [35]. Therefore, the changes in the PSFs shown in Response Fig. 14, as a result of the modulated fiber output, are very likely responsible for the 34% discrepancy between the predicted and experimental values of signal improvement in 3PF.

We would like to note that, while we only optimized the spectral band energy for the proof-of-concept experiment, optimizing PSFs together with the spectral properties is very promising in the future work since the proposed fiber shaper has demonstrated the capability of spatial control, as shown in Response Fig. 5. We are actively seeking an effective approach for multi-objective optimization to leverage the high control degrees of freedom in the fiber shaper. We have incorporated the results about the PSFs in Supplementary Figs. 12, 13.

Response Figure 14 (Added as Supplementary Figure 13): Qualitative PSF comparison of the 2PF excitation (900 nm) and the 3PF excitation (1300 nm) with the fiber source under random bending conditions induced by the fiber shaper’s different states. The size of the bead is 1 μm .

Minor comments

1. The y-axis label of Fig. 5c is misspelled as A”o”tocorrelation Signal.

2. Add a color code to the spatial intensity profiles in Fig. 5c, 6i, 7, and Supplementary Fig. 2.
3. Provide an explanation for the input wavelength in Supplementary Fig. 3-5.
4. Increase the contrast of the 3125 different states in the figures to highlight their broad tunability.
5. For “reader friendly”, at least, explain the variables A_n , β , and z in the equation (1) in the main text, even though they are explained in ref. 47.

Response: We thank the reviewer for their thorough review of our work and for the concrete suggestions. We have revised our manuscript and figures accordingly:

- We have changed the y-axis in Fig. 5c (now Fig. 4c);
- We have added color code to Fig. 5c (now Fig. 4c), Fig. 6l (now Fig. 5l), and Fig. 7 (now Fig. 6); original Fig. 6i (now Fig. 5i) presents the pseudo-colored multiphoton images, we have made this clearer in the figure caption and Methods (now Section 4.4).
- We have provided input wavelength in Supplementary Figs. 3-5 (now Supplementary Figs. 4-6).
- We have increased the contrast (now Figs. 1, 3 and Supplementary Fig. 7).
- We have revised equation (1) and added explanations to A_n and z (see Section 2.2 and Methods in the revised manuscript); β is the propagation constant and has been implicitly included in the revised $A_n(z, t)$.

References

- [1] R. T. Schermer and J. H. Cole, ‘Improved bend loss formula verified for optical fiber by simulation and experiment’, *IEEE J Quantum Electron*, vol. 43, no. 10, pp. 899–909, 2007, doi: 10.1109/JQE.2007.903364.
- [2] S. Tang, T. B. Krasieva, Z. Chen, G. Tempea, and B. J. Tromberg, ‘Effect of pulse duration on two-photon excited fluorescence and second harmonic generation in nonlinear optical microscopy’, <https://doi.org/10.1117/1.2177676>, vol. 11, no. 2, p. 020501, Mar. 2006, doi: 10.1117/1.2177676.
- [3] S. Resisi, Y. Viernik, S. M. Popoff, and Y. Bromberg, ‘Wavefront shaping in multimode fibers by transmission matrix engineering’, *APL Photonics*, vol. 5, no. 3, p. 36103, Mar. 2020, doi: 10.1063/1.5136334/570329.
- [4] N. Berti, K. Baudin, A. Fusaro, G. Millot, A. Picozzi, and J. Garnier, ‘Interplay of Thermalization and Strong Disorder: Wave Turbulence Theory, Numerical Simulations, and Experiments in Multimode Optical Fibers’, *Phys Rev Lett*, vol. 129, no. 6, p. 063901, Aug. 2022, doi: 10.1103/PHYSREVLETT.129.063901/FIGURES/4/MEDIUM.

- [5] N. K. Fontaine, R. Ryf, H. Chen, D. T. Neilson, K. Kim, and J. Carpenter, ‘Laguerre-Gaussian mode sorter’, *Nature Communications* 2019 10:1, vol. 10, no. 1, pp. 1–7, Apr. 2019, doi: 10.1038/s41467-019-09840-4.
- [6] K.-P. Ho and J. M. Kahn, ‘Optical Fiber Telecommunications VIB. 11.1 INTRODUCTION Mode Coupling and its Impact on Spatially Multiplexed Systems’, 2013, doi: 10.1016/B978-0-12-396960-6.00011-0.
- [7] W. Xiong, C. W. Hsu, Y. Bromberg, J. E. Antonio-Lopez, R. Amezcua Correa, and H. Cao, ‘Complete polarization control in multimode fibers with polarization and mode coupling’, *Light: Science & Applications* 2018 7:1, vol. 7, no. 1, pp. 1–10, Aug. 2018, doi: 10.1038/s41377-018-0047-4.
- [8] Z. Finkelstein, K. Sulimany, S. Resisi, and Y. Bromberg, ‘Spectral shaping in a multimode fiber by all-fiber modulation’, *APL Photonics*, vol. 8, no. 3, p. 36110, Mar. 2023, doi: 10.1063/5.0121539/2879063.
- [9] S. Ramachandran, J. Demas, and L. Rishøj, ‘Free-space beam shaping for precise control and conversion of modes in optical fiber’, *Optics Express*, Vol. 23, Issue 22, pp. 28531–28545, vol. 23, no. 22, pp. 28531–28545, Nov. 2015, doi: 10.1364/OE.23.028531.
- [10] M. F. Yan *et al.*, ‘Light propagation with ultralarge modal areas in optical fibers’, *Optics Letters*, Vol. 31, Issue 12, pp. 1797–1799, vol. 31, no. 12, pp. 1797–1799, Jun. 2006, doi: 10.1364/OL.31.001797.
- [11] O. Tzang, A. M. Caravaca-Aguirre, K. Wagner, and R. Piestun, ‘Adaptive wavefront shaping for controlling nonlinear multimode interactions in optical fibres’, *Nat Photonics*, vol. 12, no. 6, 2018, doi: 10.1038/s41566-018-0167-7.
- [12] B. Wetzel *et al.*, ‘Customizing supercontinuum generation via on-chip adaptive temporal pulse-splitting’, *Nat Commun*, vol. 9, no. 1, 2018, doi: 10.1038/s41467-018-07141-w.
- [13] W. Xiong, P. Ambichl, Y. Bromberg, B. Redding, S. Rotter, and H. Cao, ‘Spatiotemporal Control of Light Transmission through a Multimode Fiber with Strong Mode Coupling’, *Phys Rev Lett*, vol. 117, no. 5, 2016, doi: 10.1103/PhysRevLett.117.053901.
- [14] Z. Deng, Y. Chen, J. Liu, C. Zhao, and D. Fan, ‘Spatio-temporal control of dispersive waves trapping by solitons in graded-index multimode fibers’, *Applied Physics Express*, vol. 13, no. 11, 2020, doi: 10.35848/1882-0786/abdbb3.
- [15] M. C. Velsink, L. V. Amitonova, and P. W. H. Pinkse, ‘Spatiotemporal focusing through a multimode fiber via time-domain wavefront shaping’, *Opt Express*, vol. 29, no. 1, 2021, doi: 10.1364/oe.412714.
- [16] D. J. McCabe *et al.*, ‘Spatio-temporal focusing of an ultrafast pulse through a multiply scattering medium’, *Nat Commun*, vol. 2, no. 1, 2011, doi: 10.1038/ncomms1434.
- [17] L. G. Wright, D. N. Christodoulides, and F. W. Wise, ‘Controllable spatiotemporal nonlinear effects in multimode fibres’, *Nature Photonics* 2015 9:5, vol. 9, no. 5, pp. 306–310, Apr. 2015, doi: 10.1038/nphoton.2015.61.
- [18] M. A. Eftekhar *et al.*, ‘Versatile supercontinuum generation in parabolic multimode optical fibers’, *Opt Express*, vol. 25, no. 8, pp. 9078–9087, 2017, doi: 10.1364/OE.25.009078.

- [19] X. Wei, J. C. Jing, Y. Shen, and L. V. Wang, ‘Harnessing a multi-dimensional fibre laser using genetic wavefront shaping’, *Light Sci Appl*, vol. 9, no. 1, 2020, doi: 10.1038/s41377-020-00383-8.
- [20] U. Teğın, B. Rahmani, E. Kakkava, N. Borhani, C. Moser, and D. Psaltis, ‘Controlling spatiotemporal nonlinearities in multimode fibers with deep neural networks’, *APL Photonics*, vol. 5, no. 3, 2020, doi: 10.1063/1.5138131.
- [21] N. Bender, H. Haig, D. Christodoulides, and F. Wise, ‘Spectral Speckle Customization’, *Optica*, 2023, doi: 10.1364/optica.499461.
- [22] D. M. Mills, J. R. Helliwell, Å. Kvik, T. Ohta, I. A. Robinson, and A. Authier, ‘Report of the working group on synchrotron radiation nomenclature - Brightness, spectral brightness or brilliance?’, *J Synchrotron Radiat*, vol. 12, no. 3, 2005, doi: 10.1107/S090904950500796X.
- [23] Y. Shen *et al.*, ‘Roadmap on spatiotemporal light fields’, *Journal of Optics (United Kingdom)*, vol. 25, no. 9, 2023, doi: 10.1088/2040-8986/ace4dc.
- [24] M. A. Eftekhar, H. Lopez-Aviles, F. W. Wise, R. Amezcua-Correa, and D. N. Christodoulides, ‘General theory and observation of Cherenkov radiation induced by multimode solitons’, *Commun Phys*, vol. 4, no. 1, 2021, doi: 10.1038/s42005-021-00640-1.
- [25] L. G. Wright, W. H. Renninger, D. N. Christodoulides, and F. W. Wise, ‘Spatiotemporal dynamics of multimode optical solitons’, *Opt Express*, vol. 23, no. 3, pp. 3492–3506, 2015, doi: 10.1364/OE.23.003492.
- [26] M. Zitelli *et al.*, ‘Multimode solitons in step-index fibers’, *Opt Express*, vol. 30, no. 4, 2022, doi: 10.1364/oe.446482.
- [27] Y. Wu, N. Bender, D. N. Christodoulides, and F. W. Wise, ‘Highly multimode solitons in step-index optical fiber’, *APL Photonics*, vol. 8, no. 9, 2023, doi: 10.1063/5.0166177.
- [28] L. G. Wright *et al.*, ‘Multimode nonlinear fiber optics: Massively parallel numerical solver, tutorial, and outlook’, *IEEE Journal of Selected Topics in Quantum Electronics*, vol. 24, no. 3, 2018, doi: 10.1109/JSTQE.2017.2779749.
- [29] L. Rishøj, B. Tai, P. Kristensen, and S. Ramachandran, ‘Soliton self-mode conversion: revisiting Raman scattering of ultrashort pulses’, *Optica*, vol. 6, no. 3, 2019, doi: 10.1364/optica.6.000304.
- [30] S. Perret *et al.*, ‘Supercontinuum generation by intermodal four-wave mixing in a step-index few-mode fibre’, *APL Photonics*, vol. 4, no. 2, 2019, doi: 10.1063/1.5045645.
- [31] N. B. Terry, T. G. Alley, and T. H. Russell, ‘An explanation of SRS beam cleanup in graded-index fibers and the absence of SRS beam cleanup in step-index fibers’, *Opt Express*, vol. 15, no. 26, 2007, doi: 10.1364/oe.15.017509.
- [32] M. Ferraro *et al.*, ‘Femtosecond nonlinear losses in multimode optical fibers’, *Photonics Res*, vol. 9, no. 12, 2021, doi: 10.1364/prj.425878.
- [33] M. Ferraro *et al.*, ‘Multiphoton ionization of standard optical fibers’, *Photonics Res*, vol. 10, no. 6, 2022, doi: 10.1364/prj.451417.

- [34] T. Hansson *et al.*, ‘Nonlinear beam self-imaging and self-focusing dynamics in a GRIN multimode optical fiber: theory and experiments’, *Opt Express*, vol. 28, no. 16, 2020, doi: 10.1364/oe.398531.
- [35] C. Xu and W. W. Webb, ‘Multiphoton Excitation of Molecular Fluorophores and Nonlinear Laser Microscopy’, *Topics in Fluorescence Spectroscopy*, pp. 471–540, 2002, doi: 10.1007/0-306-47070-5_11.

REVIEWERS' COMMENTS

Reviewer #1 (Remarks to the Author):

The authors addressed all of my concerns and I appreciate their thorough response.

Reviewer #2 (Remarks to the Author):

In their response letter, the Authors have satisfactorily addressed the issues raised in my own first round of review. Concerning the Remarks of Reviewers #3, I share the concerns of this Reviewer. In their Response and in the revised manuscript, the Authors have clarified what are the merits as well as the limitations of their approach to "customize" the spectral and spatial properties of the field at the output of multimode optical fibers for improving the performances of nonlinear microscopy applications. However, I would leave to Reviewer #3 the full assessment of the Authors' response, to see if indeed they have successfully resolved all of these issues.

I have a couple of remarks for the Authors' consideration, when further revising their text:

1) In the title I would add "spatially" to "modulating", otherwise one may be (wrongly) led to think that the proposed method involves a temporal modulation of the input field.

2) I think that the information provided in response figures 2, 5, 5 and 7 is quite relevant to enrich the content of this work: I would recommend to include these figures and associated discussion in the Supplementary notes.

Reviewer #3 (Remarks to the Author):

To really prove that the exchange of energy between modes has a beneficial effect on the spatiotemporal profile using actuators, the authors would have to show not only the profile of the output mode but also the increase in energy in that mode. Otherwise, a simple linear loss on high-order modes can achieve

this spatial selection by causing losses and therefore a change in spectral broadening. But without improving the light brightness of the selected mode.

However, the authors have answered all the questions with arguments that are more or less convincing.

Reviewer #4 (Remarks to the Author):

The authors have addressed extensively my concerns.

RESPONSE TO REVIEWERS' COMMENTS (Round 2)

Reviewer #1 (Remarks to the Author):

The authors addressed all of my concerns and I appreciate their thorough response.

Reviewer #2 (Remarks to the Author):

In their response letter, the Authors have satisfactorily addressed the issues raised in my own first round of review. Concerning the Remarks of Reviewers #3, I share the concerns of this Reviewer. In their Response and in the revised manuscript, the Authors have clarified what are the merits as well as the limitations of their approach to "customize" the spectral and spatial properties of the field at the output of multimode optical fibers for improving the performances of nonlinear microscopy applications. However, I would leave to Reviewer #3 the full assessment of the Authors' response, to see if indeed they have successfully resolved all of these issues.

We appreciate the reviewer for acknowledging issues in round 1 have been satisfactorily addressed.

I have a couple of remarks for the Authors' consideration, when further revising their text:

1) In the title I would add "spatially" to "modulating", otherwise one may be (wrongly) led to think that the proposed method involves a temporal modulation of the input field.

The perturbation was applied to the multimodal nonlinear pulse propagation. Therefore, spatial modulating would be inaccurate. We believe "modulating nonlinear pulse propagation" accurately describes our approach and is distinct from the "temporal modulation of the input field."

2) I think that the information provided in response figures 2, 5, 5 and 7 is quite relevant to enrich the content of this work: I would recommend to include these figures and associated discussion in the Supplementary notes.

Figures 2 and 5 have been included as the Supplementary Figure 14 and 15 in the revised supplementary note. We did not include Response Figure 7 because we are still investigating this one and it needs more thorough characterization in follow-up work that focuses more on the interplay between wavefront control and propagation modulation.

Reviewer #3 (Remarks to the Author):

To really prove that the exchange of energy between modes has a beneficial effect on the spatiotemporal profile using actuators, the authors would have to show not only the profile of the output mode but also the increase in energy in that mode. Otherwise, a simple linear loss on high-order modes can achieve this spatial selection by causing losses and therefore a change in spectral broadening. But without improving the light brightness of the selected mode.

However, the authors have answered all the questions with arguments that are more or less convincing. We thank the reviewer for acknowledging our response has been more or less convincing. We agree that the loss of the high-order modes can lead to spatial selection and thus a change in spectral broadening. However, for the bending curvature we are operating, we have tracked the loss with different fiber configurations and they remain negligible, which makes the linear loss less of a factor in the modulation. In addition, our experiments show that we can improve light brightness of the selected mode. For example, we have shown the energy increase in different modes in Response Figure 6 in the first-round revision.

Reviewer #4 (Remarks to the Author):

The authors have addressed extensively my concerns.